# Projections from thalamic nucleus reuniens to hippocampal CA1 area participate in context fear extinction by affecting extinction-induced molecular remodeling of excitatory synapses

**Magdalena Ziółkowska, Narges Sotoudeh, Anna Cały, Monika Puchalska, Roberto Pagano[†], Malgorzata Alicja Śliwińska[‡], Ahmad Salamian, Kasia Radwanska***

Laboratory of Molecular Basis of Behavior, Nencki Institute of Experimental Biology of Polish Academy of Sciences, Warsaw, Poland

**\*For correspondence:**
k.radwanska@nencki.edu.pl

**Present address:** [†]Laboratory of Molecular and Cellular Neurobiology, International Institute of Molecular and Cell Biology, Warsaw, Poland; [‡]Electron Microscopy Platform & Bio Imaging Core at VIB-KU Leuven Center for Brain and Disease Research, and Department of Neurosciences at KU Leuven, Campus Gasthuisberg, Leuven, Belgium

**Competing interest:** The authors declare that no competing interests exist.

## eLife Assessment

This work provides **important** findings characterizing potential synaptic mechanisms supporting the role of midline thalamus-hippocampal projections in fear memory extinction in mice. The methods and approaches were considered **solid**, though some evidence is incomplete as there are some concerns with the analytical approaches used for some aspects of the study. This work will be of interest to those in the field of thalamic regulation and fear memory.

**Abstract** The ability to extinguish contextual fear in a changing environment is crucial for animal survival. Recent data support the role of the thalamic nucleus reuniens (RE) and its projections to the dorsal hippocampal CA1 area (RE→dCA1) in this process. However, it remains poorly understood how RE impacts dCA1 neurons during contextual fear extinction (CFE). Here, we reveal that the RE→dCA1 pathway contributes to the extinction of contextual fear by affecting CFE-induced molecular remodeling of excitatory synapses. Anatomical tracing and chemogenetic manipulation in mice demonstrate that RE neurons form synapses and regulate synaptic transmission in the stratum oriens (SO) and lacunosum-moleculare (SLM) of the dCA1 area, but not in the stratum radiatum (SR). We also observe CFE-specific structural changes of excitatory synapses and expression of the synaptic scaffold protein, PSD-95, in both strata innervated by RE, but not in SR. Interestingly, only the changes in SLM are specific for the dendrites innervated by RE. To further support the role of the RE→dCA1 projection in CFE, we demonstrate that brief chemogenetic inhibition of the RE→dCA1 pathway during a CFE session persistently impairs the formation of CFE memory and CFE-induced changes of PSD-95 levels in SLM. Thus, our data indicate that RE participates in CFE by regulating CFE-induced molecular remodeling of dCA1 synapses.

## Introduction

The distinction between dangerous and safe contexts, as well as the ability to update such information in a changing environment, is crucial for animal survival as mistakes can have costly consequences. Impairments in context-dependent behaviors have been associated with pathological conditions such as post-traumatic stress disorder, schizophrenia, and substance use disorders (*Lissek et al., 2014;*

*Garfinkel et al., 2014*; *Grodin et al., 2018*; *Maren et al., 2013*). Therefore, understanding the molecular and cellular mechanisms that underlie the updating of contextual memories is crucial for developing new therapeutic approaches to alleviate persistent and unmalleable contextual fear or compulsive context-induced drug seeking (*Ji and Maren, 2007*; *Maren and Quirk, 2004*; *Myers and Davis, 2002*).

One way to study contextual memories in the laboratory animals is through contextual fear conditioning (CFC) (*Maren et al., 2013*). In CFC, a spatial context, referred to as the conditioned stimulus (CS), is repeatedly paired with a noxious unconditioned stimulus (US), typically a mild shock. Freezing behavior serves as a quantifiable measure of contextual fear memory. With continued exposure to the CS in the absence of the US, freezing behavior generally decreases and exploratory behavior increases. This process is termed contextual fear extinction (CFE) and occurs as animals learn over time that the context no longer predicts shocks. For extinction to take place, mice must actively recall and update contextual memories. Therefore, mechanisms must exist in the brain to allow for these processes.

The classic neuronal circuit of CFE includes medial prefrontal cortex (mPFC), amygdala, and hippocampus (*Maren et al., 2013*; *Orsini and Maren, 2012*). However, recent data also support the role of the thalamus in this process (*Venkataraman and Dias, 2023*; *Cassel et al., 2013*). In particular, nucleus reuniens (RE), a ventral midline thalamic nucleus, sends axon collaterals to mPFC and the hippocampal CA1 area (*Hoover and Vertes, 2012*; *Varela et al., 2014*; *Vertes et al., 2007*; *Goswamee et al., 2021*; *Dolleman-Van der Weel et al., 1997*; *Wouterlood et al., 1990*), forming a disynaptic link that can coordinate neural activity between the two structures (*Jayachandran et al., 2023*; *Dolleman-van der Weel et al., 2019*; *Griffin, 2021*; *Totty et al., 2023*). Activity-dependent brain mapping studies have revealed increased cfos expression in the RE following CFE learning and recall (*Ramanathan et al., 2018*; *Silva et al., 2019*). Activity of RE is synchronized with freezing bouts (*Silva et al., 2021*; *Ratigan et al., 2023*), suggesting that activity patterns in the RE are related to the suppression of fear responses. Indeed, muscimol-induced inactivation of RE following a weak fear conditioning procedure enhances fear memory consolidation (*Troyner et al., 2018*). Moreover, inactivation of the RE prior to extinction training prevents extinction learning, while RE inactivation prior to retrieval impairs retrieval of the extinction memories (*Ramanathan et al., 2018*). Finally, inactivation of the RE also impairs the precision of contextual fear memories and results in fear generalization (*Ramanathan et al., 2018*; *Troyner et al., 2018*; *Xu and Südhof, 2013*), a process that requires both the HPC and mPFC (*Rozeske et al., 2015*; *Zelikowsky et al., 2014*). Together, these studies suggest that RE is critical for mPFC-HPC synchrony during the retrieval and updating of contextual memories. This hypothesis is supported by recent studies linking RE→dCA1 activity with recall of fear and extinction memory (*Totty et al., 2023*; *Ratigan et al., 2023*). Still, it remains unknown how RE impacts the dCA1 function to participate in CFE.

Here, we sought to determine whether the RE→dCA1 projection plays a role in updating context memories and characterize underlying synaptic processes. First, we used chemogenetic manipulations (*Armbruster et al., 2007*) and conducted ex vivo field recordings in mice to analyze the impact of the RE afferents on synaptic transmission in three strata of the hippocampal CA1 area: the stratum oriens (SO), radiatum (SR), and lacunosum-moleculare (SLM). Secondly, we employed virally mediated labeling of RE afferents in Thy1-GFP mice (*Feng et al., 2000*) to analyze structural changes of the RE boutons and dendritic spines (as a proxy for glutamatergic synapses) in dCA1 during contextual fear memory extinction. This approach was complemented by serial block-face scanning electron microscopy (SBFSEM) (*Denk and Horstmann, 2004*; *Śliwińska et al., 2021*) to analyze structural synaptic changes induced by CFE with nanoscale resolution. Finally, using chemogenetic manipulations, we tested the role of the RE in CFE-induced synaptic changes in dCA1, and the role of RE→dCA1 projection in the CFE. Overall, our data support the notion that the RE→dCA1 pathway contributes to the extinction of contextual fear by affecting CFE-induced molecular synaptic changes in dCA1.

## Results

### Assessment of projections from RE to dorsal CA1 area (RE→dCA1)

To examine RE axons in dCA1 (RE→dCA1), mice were stereotaxically injected into RE with adeno-associated viral vector (isotype 2.1) encoding red fluorescent protein, mCherry, under *camk2a*

promoter (AAV$_{2.1}$:camk2a_mCherry) (*Figure 1A*), resulting in mCherry expression in more than 60% of RE cells spreading from Bregma –0.46 mm to –1.5 mm (*Figure 1B*). RE axons labeled by mCherry were analyzed in the SO, SLM, and SR of dCA1 (*Figure 1C and D*). We analyzed 21 tissue bricks (67.7 × 67.7 × 5.4 μm) per stratum, containing on average 37 μm of axons per tissue bricks in SO, 0.7 μm in SR, and 65 μm in SLM. RE boutons were identified in SO and SLM as axonal thickenings in close apposition to PSD-95-positive puncta (a synaptic scaffold used as a marker of excitatory synapses that can be located both on excitatory and inhibitory neurons; *Chen et al., 2011*; *El-Husseini et al., 2000*; *Kornau et al., 1995*; *Dharmasri et al., 2024*; *Figure 1D*). Density of RE boutons in SLM was slightly lower, and boutons were much larger, compared to SO boutons (*Figure 1F and G*). As RE axonal fragments were very short and rare in SR, we did not analyze their boutons.

To test the function of RE→dCA1 projections, mice were stereotaxically injected with AAV$_{2.1}$ expressing double-floxed inverted open-reading frame (DIO) of Gi-protein-coupled inhibitory DREADD receptor (hM4Di) and mCherry under human synapsin promoter (AAV$_{2.1}$:hSyn_DIO_hM4Di _ mCherry) into RE allowing for Cre-dependent expression of hM4Di, and with canine adenoviral vector 2 (CAV$_2$) encoding Cre recombinase with GFP under *CMV* promoter (CAV$_2$:CRE_GFP) bilaterally into dCA1 allowing for retrograde targeting of neurons innervating dCA1 (*Figure 1H*). This combinatorial manipulation enabled hM4Di expression specifically in RE neurons projecting to dCA1 (RE→dCA1) (*Figure 1I*). It is, however, likely that some of these neurons bifurcate and also innervate PFC (*Hoover and Vertes, 2012*). Mice received i.p. injection of saline or hM4Di agonist (CNO, 3 mg/kg) and were sacrificed 30 min later. Their brains were sliced and used for recording field excitatory postsynaptic potentials (fEPSP) and fiber volley (FV) in SO, SR, and SLM of dCA1, while axons in these strata were stimulated (*Figure 1J–L*). CNO decreased fEPSP in SO and SLM, and tended to decrease FV in SLM. No effect of CNO on these synaptic measures was observed in SR. Hence, our data indicate that RE innervates and efficiently regulates excitatory synaptic transmission in SLM and SO.

## Analysis of dCA1 synapses after CFE

dCA1, RE, as well as RE→dCA1 projections have been implicated in the extinction of contextual fear (*Venkataraman and Dias, 2023*; *Cassel et al., 2013*; *Totty et al., 2023*; *Ratigan et al., 2023*). Moreover, our recent study showed that the extinction of contextual fear remodels dendritic spines specifically in SO and SLM of dCA1 (*Ziółkowska et al., 2023*), two strata innervated by RE (*Figure 1E*). Hence, next we tested whether CFE-induced structural plasticity of dCA1 synapses is specific for the dendrites innervated by RE. To this end, Thy1-GFP(M) mice with sparse GFP expression in the excitatory neurons (*Feng et al., 2000*) were stereotaxically injected with AAV$_{2.1}$:camk2a_mCherry into RE. This manipulation allowed us to analyze both pre- and postsynaptic elements in the RE→SLM synapses (*Figures 2 and 3*, respectively). Three weeks after the surgery, mice underwent CFC with five electric shocks as US in novel context and 24 hr later they were re-exposed to the same context for 30 min without US presentation for CFE. At the beginning of the CFC, freezing levels were low but increased during the training. At the beginning of the CFE session, freezing levels were high, indicating contextual fear memory, and decreased within the session, indicating formation of CFE memory (*Figure 2A*; *Ziółkowska et al., 2023*). One group of mice was sacrificed 24 hr after CFC (5US) and one was re-exposed to the training context for CFE and sacrificed immediately after the session (Ext) (*Figure 2A*). In addition, the control animals were taken from home cages (Naive). Post-training analysis of the brain sections showed that mCherry was expressed in RE, the levels of AAV transduction did not differ between the experimental groups, and was observed in over 68% of RE cells (*Figure 2B*).

The analysis of the presynaptic boutons in SO revealed no significant differences in density and size between the experimental groups (*Figure 2C*). Similarly in SLM, the density of RE boutons did not differ between the experimental groups. However, the Ext mice had significantly larger RE boutons compared to the 5US animals, and no difference was observed between the Ext and Naive mice (*Figure 2D*). Hence, our data suggest that the behavioral training did not affect RE→SO presynapses, while CFC decreased the size of RE→SLM pre-synapses and this process was reversed by CFE.

To check if the presynaptic changes were coupled with postsynaptic alterations, we analyzed dCA1 dendritic spines in the same mice. We distinguished RE+ dendrites (with at least one dendritic spine in close proximity to RE boutons), RE+/+ dendritic spines (in close proximity to RE bouton), RE+/- spines (located on RE+ dendrites but without detected contact with RE boutons) and RE- dendrites and spines (that did not colocalize with RE boutons within the analyzed tissue brick) (*Figure 3A*). The

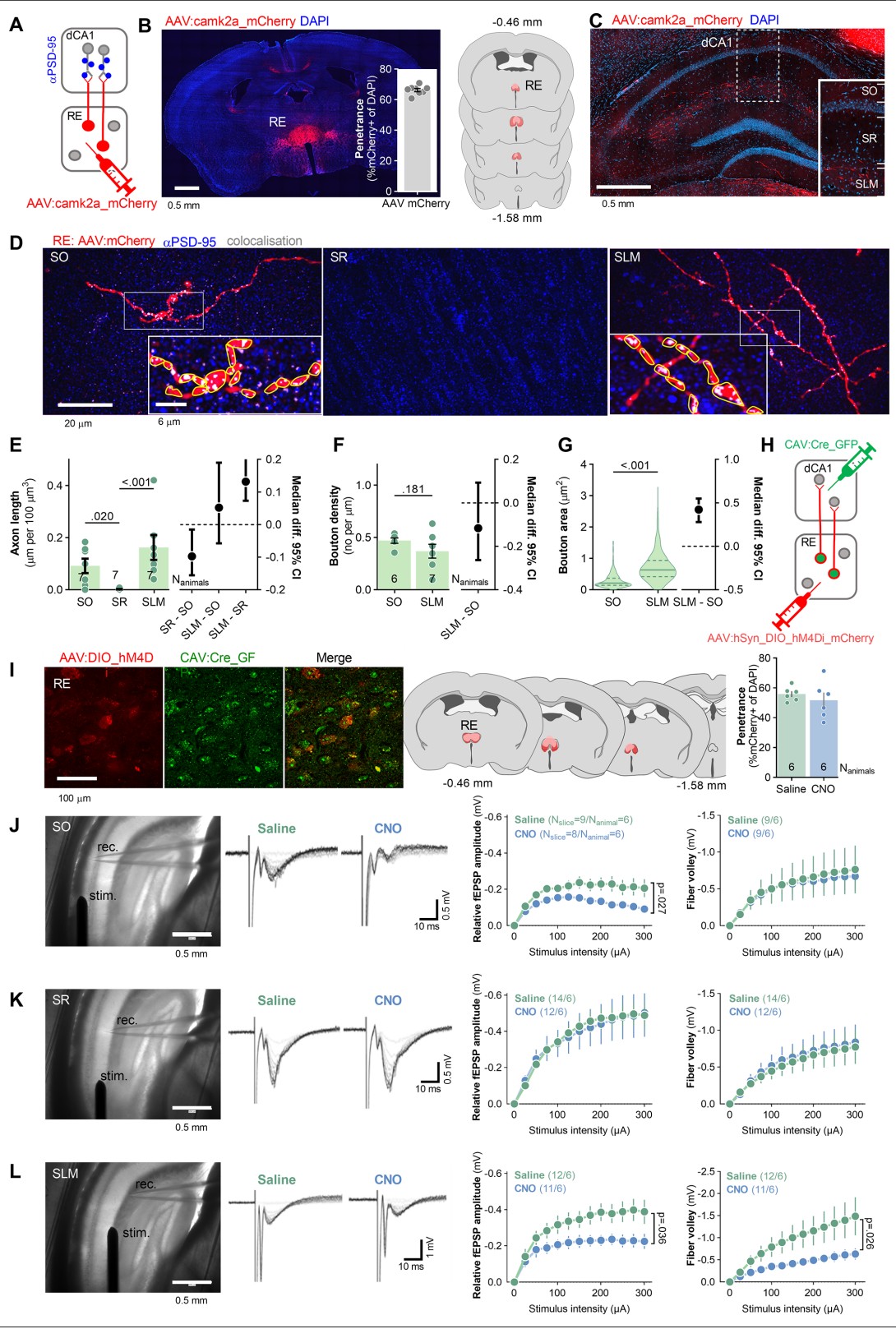

**Figure 1.** Nucleus reuniens (RE) innervates stratum oriens (SO) and stratum lacunosum-moleculare (SLM) of dCA1. (**A, B**) Experimental schema and virus expression. (**A**) Mice (N$_{animals}$ = 7) were stereotaxically injected into RE with AAV$_{2.1}$ encoding mCherry. (**B**) A microphotography of mCherry expression in RE, virus penetrance, and extent of viral expression (pink, minimal; red, maximal). (**C, D**) Microphotographs showing RE axonal fibers (red), PSD-95 immunostaining (blue), and axon colocalization with PSD-95 (white) in dorsal hippocampus (**C**) and dCA1 strata (**D**): SO, stratum radiatum, (SR) and SLM.

*Figure 1 continued on next page*

*Figure 1 continued*

RE boutons (red) were identified as axon thickenings colocalizing with PSD-95-positive puncta (gray). In insets, identified boutons outlined (yellow) in ImageJ. (**E–G**) Summary of results showing RE axon length Mann–Whitney test, $U$ = 11.5, p<0.001; effect size, SR-SO: −0.0974 [95CI 0.0152; −0.155]; SLM-SO: −0.0518 [95CI 0.190,−0.057]; SLM-SO: 0.131 [95CI 0.207; 0.0732], bouton density (Mann–Whitney test, $U$ = 108, p=0.060; $N_{animals}$ = 7; effect size, SLM-SO: −0.137 [95CI 0.004; −0.244]) and bouton area in dCA1 strata (Mann–Whitney test, $U$ = 12,550. p<0.001; SO $N_{boutons}$ = 266; SLM $N_{boutons}$ = 282; effect size: SLM-SO, 0.367 [95CI 0.419; 0.321]). (**E**) Each dot represents one tissue brick. (**F**) Each dot represents one axon. (**G**) Violin plots are based on data for individual boutons. (**H–L**) The analysis of synaptic responses in dCA1 in mice after chemogenetic inhibition of RE→dCA1. (**H**) Experimental schema. Mice were stereotaxically injected with $CAV_2$ encoding Cre-recombinase with green fluorescent protein (CAV₂:Cre_GFP) into dCA1 and with $AAV_{2.1}$ encoding Cre-dependent inhibitory DREADD with mCherry (AAV:DIO_hM4Di _mCherry) into RE, resulting in the expression of Cre in neurons projecting to dCA1, and hM4Di in dCA1-projecting neurons of RE (RE→dCA1). Mice were i.p. injected with saline or CNO (3 mg/kg) and sacrificed 30 min later. Their brains were sliced and used for electrophysiological analysis of dCA1 synaptic responses. (**I**) Microphotographs showing hM4Di expression in RE→dCA1 neurons, extent of viral expression (pink, minimal; red, maximal), and virus penetrance in RE. (**J–L**) Microphotographs of the experimental setups (stim., stimulating electrode; rec., recording electrode), representative fEPSPs traces, summary of data for input–output plots of fEPSP slopes and fiber volley (FV) amplitude recorded in SO (**J**), SR (**K**) and SLM (**L**) in response to increasing intensities of stimulation. CNO decreased fEPSP in SLM (RM ANOVA, effect of CNO, F(1, 24)=4.95, p=0.036) and SO (F(1, 16)=5.906, p=0.027) but not in SR (F(1, 20)=0.681, p=0.419). CNO slightly decreased FV amplitudes in SLM (RM two-way ANOVA, F(1, 19)=3.752, p=0.068) but not SO (F(1, 10)=0.020, p=0.888) or SR (F(1, 25)=0.042, p=0.839). **E, F, J-L** graphs show means +/- SEM. G graph shows medians +/- IQR.

average length of the analyzed dendritic fragments per tissue brick (67.7 × 67.7 × 5.4 μm) was 50 μm in SO, 53 in SR, and 40 μm in SLM. They contained on average respectively 25, 37, and 32 dendritic spines, including 1–3 RE+/+ in SO and 1–4 RE+/+ in SLM. The analysis of SO found that the density and size of dendritic spines in the Ext group did not differ from those analyzed in the Naive animals. However, the 5US animals tended to have more spines, and these spines were smaller and contained less PSD-95 as compared to those in the Naive and Ext animals (*Figure 3B*, *Figure 3—figure supplement 1A–C*, and *Supplementary file 1a–c*). Interestingly, these spine changes were specific for the RE- dendrites. Hence, CFE reversed synaptic changes on RE- dendrites induced by CFC. On the other hand, we did not observe any CFE-specific changes of RE+/- and RE+/+ spines. However, RE+/- spines in 5US and Ext groups contained less PSD-95 compared to the Naive animals. Hence, our observation confirmed our former findings of CFE-induced synaptic plasticity in SO (*Ziółkowska et al., 2023*) and specified that these global changes result mostly from the alterations of the RE- dendrites. These observations also match the lack of CFE-induced changes on RE buttons in SO (*Figure 2C*).

In SR, we did not observe any significant training-induced changes in density and size of dendritic spines. However, PSD-95 levels per dendritic spine were decreased in the Ext and 5US groups compared to the Naive animals (*Figure 3C*, *Figure 3—figure supplement 1D–F*, and *Supplementary file 1d–f*).

In SLM, no significant changes in dendritic spine density were observed globally or on RE- dendrites; however, we found increased density of RE+/+ dendritic spines (*Figure 3D*, *Figure 3—figure supplement 1G–I*, and *Supplementary file 1g–i*). There were also more RE+/+ spines in the Ext group compared to the 5US animals. Moreover, the median dendritic spine area and PSD-95 expression per dendritic spine were higher in the Ext group compared to 5US animals, and these differences were specific for the RE+ dendrites. Again, no such changes were observed on the RE- dendrites. Hence, in SLM, CFE resulted in synaptogenesis of RE+/+ spines, growth of RE+/+ and RE+/- spines, and accumulation of PSD-95 in RE+/- spines, supporting the idea of CFE-induced strengthening of RE→SLM projections. These observations match the CFE-induced growth of RE buttons in SLM (*Figure 2D*).

To test whether the observed changes of synaptic boutons and dendritic spines in SLM were specific for CFE, we conducted a control experiment with Thy1-GFP mice expressing mCherry in RE neurons (*Figure 4A*). The animals were exposed to the novel experimental context without electric shocks and were sacrificed 24 hr later (Ctx) or immediately after re-exposure to the same context for 30 min (Ctx-Ctx). The mice showed low levels of freezing both during the first and second exposure to the experimental context, indicating no fear memory. Post-training analysis of the brain sections revealed no significant difference in AAV penetrance in RE between the experimental groups (penetrance: Ctx: 66%; Ctx-Ctx: 60%). We also observed no global differences in density and size of RE boutons (*Figure 4B and C*) and dendritic spines (*Figure 4D and E*) in SLM between the Ctx and Ctx-Ctx animals.

Overall, our data demonstrate that synapses in all dCA1 strata undergo structural or molecular changes relevant to CFC and/or CFE. However, only in SLM CFE-induced synaptic changes are likely

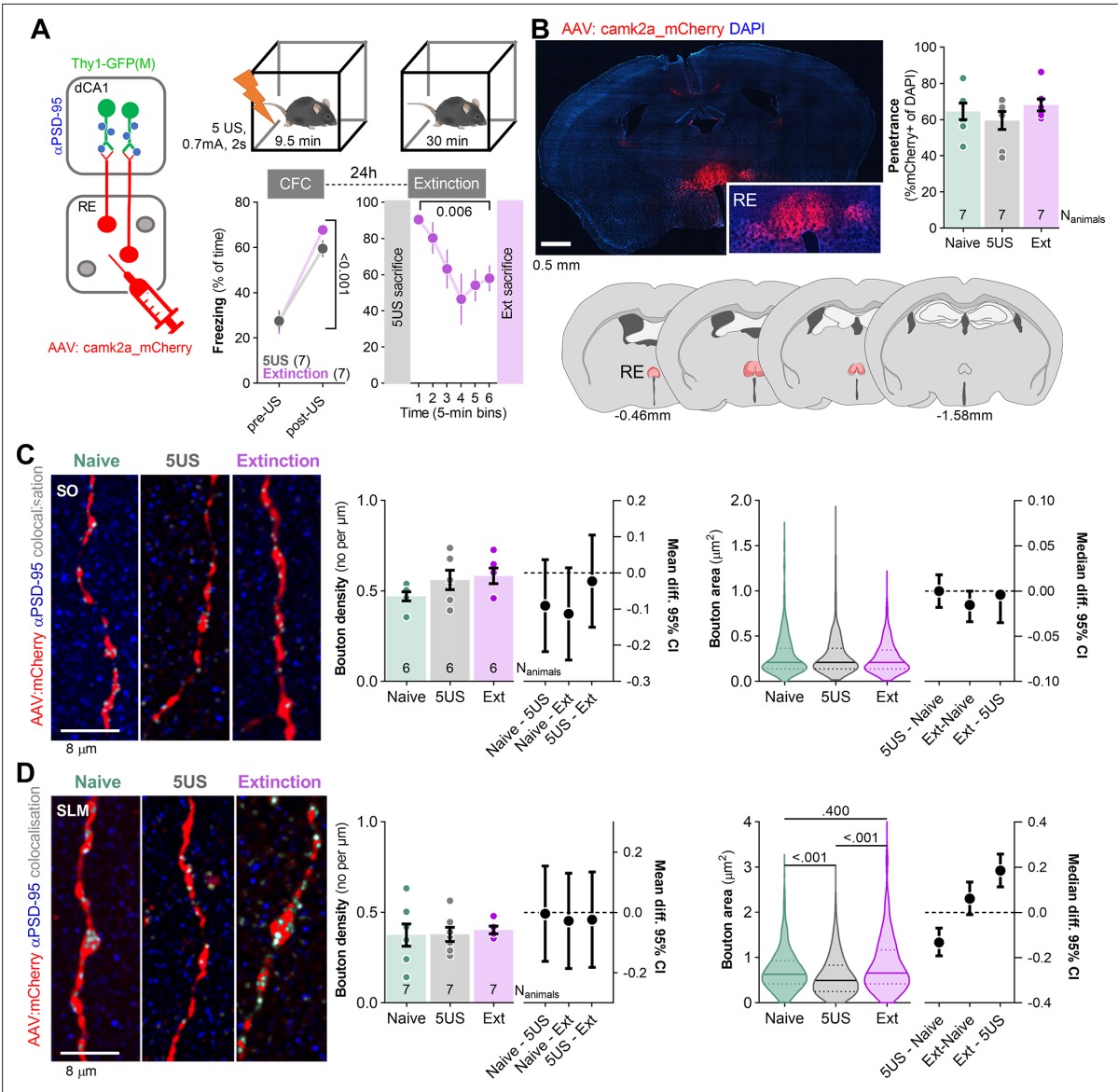

**Figure 2.** Extinction of contextual fear remodels nucleus reuniens (RE) boutons in stratum lacunosum-moleculare (SLM). (**A**) Experimental timeline and summary of data showing freezing levels during contextual fear conditioning (CFC) and contextual fear extinction (CFE) session. After injection of $AAV_{2.1}$ encoding mCherry into RE, Thy1-GFP(M) mice underwent CFC and were sacrificed 24 hr later (5US) or underwent CFE session and were sacrificed after the session (Ext) (RM ANOVA with post-hoc LSD test, effect of time: $F_{(2.74, 10.9)}=8.25$, p=0.004). Naive animals were used as a control group. (**B**) Summary of mCherry expression analysis in RE. Confocal scan of RE with local expression of mCherry, penetrance of mCherry, and extent of viral expression (pink, minimal; red, maximal). Means +/- SEM are shown. (**C, D**) RE axons analysis in dCA1. (**C**) Representative images of RE axons, immunostaining for PSD-95 protein, and their colocalization in SO (left) and summary of data showing bouton density (Mann–Whitney test, $U = 1.73$; p=0.443, means +/- SEM are shown; effect size, Naive - 5US: −0.0836 [95CI 0.0644; −0.2231]; Naive - Ext: −0.077 [95CI 0.0643; −0.2185]; 5US - Ext: 0.0065 [95CI 0.1457; −0.1326]), and bouton area (Mann–Whitney test, $U = 0.096$; p=0.960, medians +/- IQR are shown; effect size, 5US - Naive: 0.0 [95CI 0.018; −0.018]; Ext - Naive: −0.0152 [95CI 0.0; −0.034]; Ext - 5US: −0.0040 [95CI 0.0; −0.035]; Naive $N_{boutons} = 257$, $N_{animals} = 6$; 5US $N_{boutons} = 203$, $N_{animals} = 6$; Extinction $N_{boutons} = 400$, $N_{animals} = 7$) (right). (**D**) Representative images of RE axons, immunostaining for PSD-95 protein, and their colocalization in SLM (left) and data analyzing of bouton density (unpaired $t$-test, $t_{(46)} = 0.642$; p=0.523, means +/- SEM are shown; effect size, Naive - 5US: −0.0331 [95CI 0.1245; −0.1908]; Naive - Ext: −0.0054 [95CI 0.1631; −0.1522]; 5US - Ext: 0.0386 [95CI 0.1889; −0.1117]) and bouton area (Mann–Whitney test, $U = 29409$; p<0.001, medians +/- IQR are shown; effect size, Naive - 5US: 0.1854 [95CI 0.2585; 0.1137]; Naive - Ext: 0.0614 [95CI 0.1346; −0.0097]; 5US - Ext: −0.132 [95CI −0.069; −0.192]; Naive $N_{boutons} = 278$, $N_{animals} = 7$; 5US $N_{boutons} = 272$, $N_{animals} = 7$; Extinction $N_{boutons} = 291$, $N_{animals} = 7$) (right).

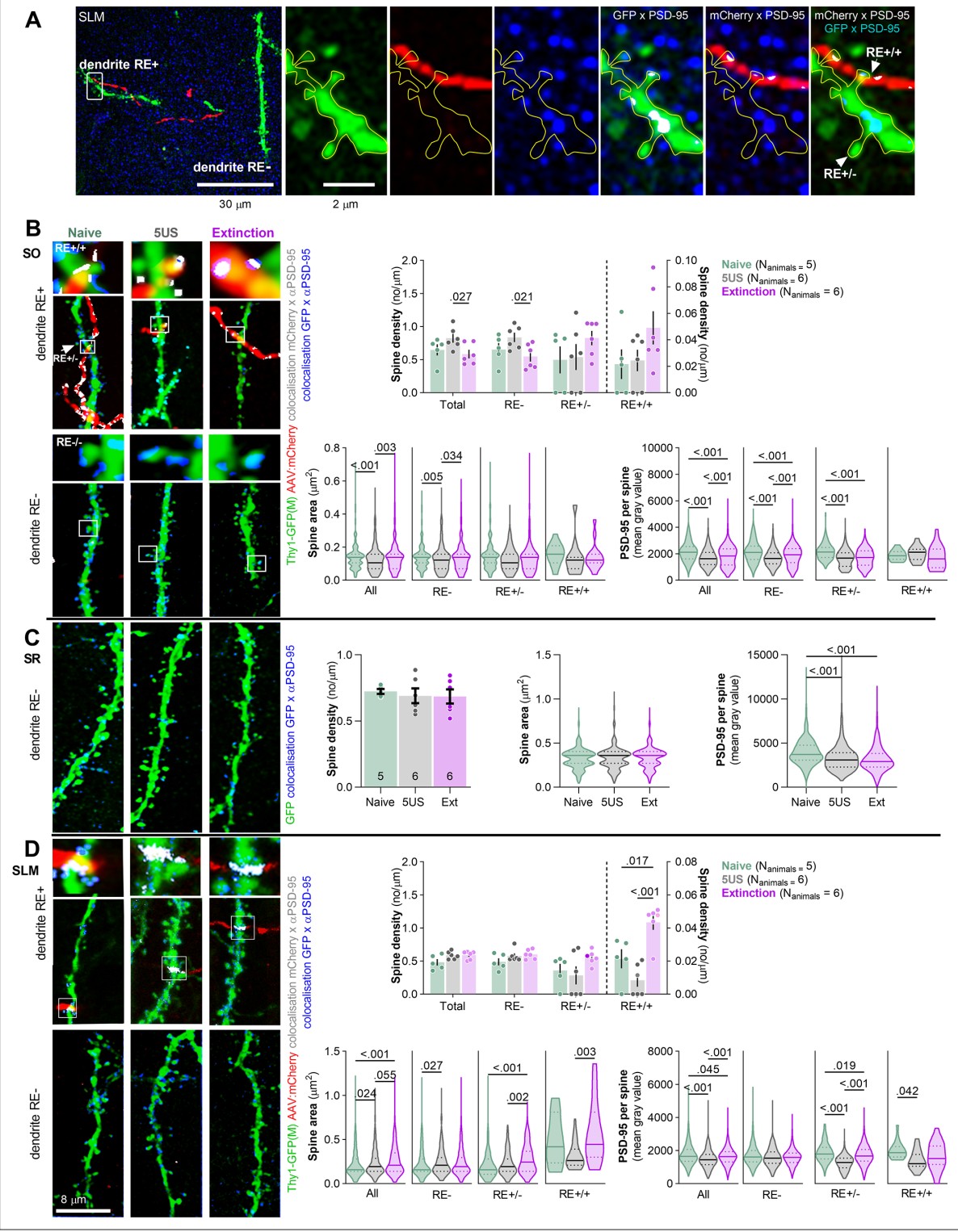

**Figure 3.** Extinction of contextual fear remodels stratum lacunosum-moleculare (SLM) dendritic spines on nucleus reuniens (RE)+ dendrites. Thy1-GFP mice were injected with AAV$_{2.1}$:camk2a_mCherry into RE and 21 days later they underwent contextual fear conditioning (CFC) and were sacrificed 24 hr later (5US, n = 6), or underwent contextual fear extinction session and were sacrificed after the session (Extinction, n = 6) (see *Figure 2A*). Naive animals (n = 5) were used as a control. For the estimation statistics, see *Figure 3—figure supplement 1*, *Supplementary file 1*. (**A**) Representative high-magnification scans in SLM showing RE axons (mCherry), dendritic spines (GFP), immunostaining for PSD-95 protein (blue), axon × PSD-95 colocalizations (gray), and dendrite × PSD-95 colocalizations (gray). Colocalizations were identified on individual confocal planes

*Figure 3 continued on next page*

*Figure 3 continued*

using the Colocalization Highlighter function (ImageJ), and next transformed into max projections. (**B**) Analysis of dendritic spines in stratum oriens (SO) . (Left) Representative high-resolution images of RE axons (red), RE+ and RE- dendritic fragments (green), colocalization of dendrites with PSD-95 immunostaining (blue), and axons with PSD-95 (gray) in SO. Insets show magnification of RE+ spines. (Rright) Summary of data showing density of dendritic spines (two-way ANOVA with Šídák's multiple comparisons test, effect of training: $F_{(2, 14)}=0.526$, p=0.602; effect of spine type: $F_{(1,39, 19,1)}=33.3$, p<0.001; spine type × training interaction: $F_{(6, 41)}=2.54$, p=0.035; density of RE+/+ spines is plotted on the right Y-axis), dendritic spine area (Kruskal–Wallis test with Dunn's multiple comparisons test, All: $U = 14.8$; p<0.001; RE-: $U = 11.1$; p=0.004; RE+/-: $U = 9.54$; p=0.008; RE+/+: $U = 3.28$; p=0.191) and PSD-95 expression per dendritic spine (Kruskal–Wallis test with Dunn's post hoc tests for planned comparisons, All: $U = 116$; p<0.001; RE-: $U = 92.03$; p<0.001; RE+/-: $U = 57.33$; p<0.001; RE+/+: $U = 1.44$; p=0.487; Naive, RE+/-: $N_{spines} = 172$; RE+/+: $N_{spines} = 15$; RE-: $N_{spines} = 407$; 5US, RE+/-: $N_{spines} = 149$; RE+/+: $N_{spines} = 12$; RE-: $N_{spines} = 581$; Extinction, RE+/-: $N_{spines} = 302$; RE+/+: $N_{spines} = 16$; RE-: $N_{spines} = 949$). (**C**) Analysis of dendritic spines in stratum radiatum (SR). Representative high-resolution images of RE- dendritic fragments (green), and colocalization of GFP with PSD-95 immunostaining (blue). Summary of data showing density of dendritic spines (one-way ANOVA, $F_{(2, 15)}=0,0085$, p=0.991), dendritic spine area (Kruskal–Wallis test, $U = 3.51$; p=0.173) and PSD-95 expression per dendritic spine (Kruskal–Wallis test, $U = 154$; p<0.001). Naive $N_{spines} = 648$; 5US $N_{spines} = 643$; Extinction $N_{spines} = 715$. (**D**) Analysis of dendritic spines in stratum lacunosum-moleculare (SLM). (Left) Representative high-resolution images of RE axons (red), RE+ and RE- dendritic fragments (green), colocalization of dendrites with PSD-95 immunostaining (blue), and axons with PSD-95 (gray). (Right) Summary of data showing density of dendritic spines (two-way ANOVA with LSD post hoc test, effect of training: $F_{(2, 13)}=2.60$, p=0.113; effect of spine type: $F_{(3, 37)}=58.3$, p<0.001, spine type × training interaction: $F_{(6, 37)}=1.34$, p=0.265; density of RE+/+ spines is plotted on the right Y-axis), dendritic spine area (Kruskal–Wallis test with Dunn's multiple comparisons test, All: $U = 28$; p<0.001; RE-: $U = 8.34$; p=0.015; RE+/-: $U = 33$; p<0.001; RE+/+: $U = 11.3$; p=0.004) and PSD-95 expression per dendritic spine (Kruskal–Wallis test with Dunn's multiple comparisons test, All: $U = 72.3$, p<0.001; RE-: $U = 5.92$; p=0.052; RE+/-: $U = 140$; p<0.001; RE+/+: $U = 6.50$; p=0.039). Naive, RE+/-: $N_{spines} = 182$; RE+/+: $N_{spines} = 16$; RE-: $N_{spines} = 524$; 5US, RE+/-: $N_{spines} = 225$; RE+/+: $N_{spines} = 15$; RE-: $N_{spines} = 424$; Extinction, RE+/-: $N_{spines} = 357$; RE+/+: $N_{spines} = 27$; RE-: $N_{spines} = 541$. Spine densities are shown as means +/- SEM. Spine area and PSD-95 per spine data is shown as medians +/- IQR.

The online version of this article includes the following figure supplement(s) for figure 3:

**Figure supplement 1.** Extinction of contextual fear remodels stratum lacunosum-moleculare (SLM) dendritic spines on nucleus reuniens (RE)+ dendrites (estimation statistics).

to be directly regulated by RE inputs as they appear on RE+ dendrites and spines. Since such changes of SLM synapses were not observed in the animals re-exposed to the neutral context, our data support the role of the described structural plasticity at the RE→SLM synapses in CFE, rather than in processing contextual information in general. Moreover, as some CFE-specific synaptic changes spread on RE+ dendrite beyond RE+ synapses, our data suggest a more global impact of RE on synaptic plasticity in dCA1. It is, however, important to mention that the analysis of RE- spines and dendrites should be approached with caution due to methodological limitations. We assume that the identification of RE-dendritic spines had only 68% accuracy due to 68% penetrance of mCherry AAV in RE (*Figure 2B*).

## Nanoscale resolution analysis of SLM synapses after CFE

Next, to verify whether CFE induces the growth of SLM synapses, we utilized SBFSEM. Although the SBFSEM approach lacks specificity for RE→SLM projections, we chose this technique primarily because it allowed us to reconstruct SLM dendritic spines with nano-scale resolution, as well as to image and directly analyze postsynaptic densities (PSDs) representing the postsynaptic part of the excitatory synapse (*Figure 5A and B*), rather than rely on the growth of dendritic spines and presence of synaptic markers that are imperfect indicators of synapses. Additionally, the CFE-induced increase in dendritic spine area and PSD-95+ puncta intensity was observed not only when analyzing RE+ dendrites separately but also when RE+ dendrites were pooled with RE- (All) (*Figure 3C*). Hence, we assumed that the overall growth of SLM synapses should be detectable also with SBFSEM as a consequence of the large structural and molecular alterations of RE+ dendrites that affected the global statistics.

In total, we reconstructed 238 dendritic spines with PSDs from the brains of the mice sacrificed 24 hr after CFC (5US, n = 5), 257 of the mice sacrificed after CFE session (Ext, n = 6), and 315 of the naive mice (Naive, n = 4). We found that dendritic spine density did not differ between the experimental groups (*Figure 5C*). However, dendritic spine volume, PSDs volume, and surface area were bigger in the Ext group compared to the Naive and 5US animals (*Figure 5C–E*). Moreover, we analyzed the correlation between dendritic spine volume, PSD volume, and PSD surface area (*Figure 5D and E*). These parameters were tightly correlated in all experimental groups. However, the lines describing the correlations in the Ext animals were slightly moved upward compared to the 5US group, indicating that during the CFE session PSDs increased in size more than dendritic spines.

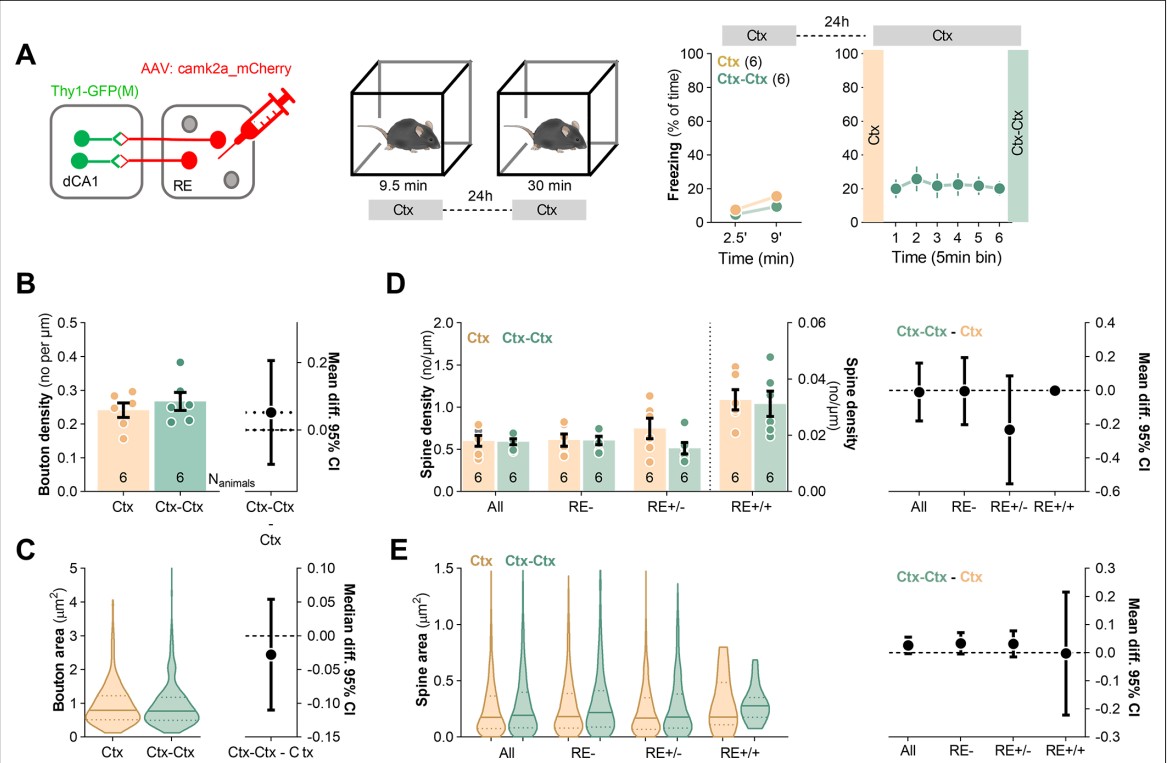

**Figure 4.** Repeated exposure to novel context exposure neither axonal boutons nor dendritic spines in stratum lacunosum-moleculare (SLM). (**A**) Experimental timeline and summary of data showing mice activity levels during novel context exposure. Thy1-GFP mice were stereotaxically injected with AAV$_{2.1}$:camk2_mCherry into nucleus reuniens (RE) and 21 days later they were exposed to a novel context. Mice were sacrificed 24 hr after novel context exposure (Ctx, N$_{animals}$ = 6) or they were re-exposed to the context and sacrificed immediately afterward (Ctx-Ctx, N$_{animals}$ = 6). (**B, C**) Morphological analysis of RE axons in stratum lacunosum-moleculare (SLM). (**B**) Summary of data showing bouton density (unpaired $t$-test, $t(10)$ = 0.755, p=0.468; effect size for Ctx-Ctx - Ctx: −0.069 [95CI 0.0351; −0.139]) and (**C**) bouton area (Mann–Whitney test, $U$ = 2745; p=0.48; effect size, Ctx-Ctx - Ctx: −0.028 [95CI 0.054; −0.110]). Ctx: N$_{boutons}$ = 230, N$_{animals}$ = 6; Ctx-Ctx: N$_{boutons}$ = 239, N$_{animals}$ = 6. (**D, E**) Analysis of dendritic spines in SLM. (**D**) Summary of data showing spine density (two-way ANOVA with Šídák's multiple comparisons test, effect of training: F(1, 20)=1.29, p=0.270, effect of spine type: F(1.91, 12.8)=53.1, p<0.001; effect size for Ctx-Ctx - Ctx, All: −0.071 [95CI 0.044; −0.186]; RE-: −0.062 [95CI 0.099; −0.223]; RE+/-: −0.182 [95CI −0.033; −0.330]; RE+/+: 0.007 [95CI 0.211; −0.197]) and (**E**) spine area (Kruskal–Wallis tests Dunn's multiple comparisons test; $U$ = 19.9, p=0.004; effect size for Ctx-Ctx - Ctx, All: 0.027 [95CI 0.056; −0.001]; RE-: 0.030 [95CI 0.077; −0.015]; RE+/-: 0.035 [95CI 0.072; −0.002]; RE+/+:−0.003 [95CI 0.216; −0.222]; Ctx RE-: N$_{spines}$ = 458, RE+/-: N$_{spines}$ = 590, RE+/+: N$_{spines}$ = 17; Ctx-Ctx RE-: N$_{spines}$ = 338, RE+/-: N$_{spines}$ = 589, RE+/+: N$_{spines}$ = 18). For (**A, B, D**), data are presented as means and SEM. For (**C, E**), data are presented as medians and IQR.

In particular, these differences seemed to occur among small dendritic spines (volume < 0.04 μm³). To test this hypothesis, we analyzed the density and volume of macular PSDs, which are usually small, and complex PSDs, which are usually large (*Figure 5F and G*; *Borczyk et al., 2021*). Neither density of macular nor complex PSDs differed between the experimental groups. However, the volume of macular, but not complex, PSDs in the Ext group was significantly larger than in the 5US animals. The volumes of PSDs of both categories were larger in the Ext group compared to naive animals.

Overall, SBFSEM analysis confirmed that CFE results in the growth of SLM dendritic spines and excitatory synapses located on the spines, suggesting their strengthening (*Holtmaat and Svoboda, 2009*; *Patterson and Yasuda, 2011*). In particular, we observed growth in the category of small macular PSDs, possibly explaining how such synaptic changes could occur so rapidly. No volume changes were observed in complex PSDs in the Ext groups compared to the 5US animals possibly because large synapses are more stable (*Borczyk et al., 2021*). Importantly, our correlation analysis for PSDs and dendritic spines demonstrates that the analysis of dendritic spine volumes, as a proxy of synaptic changes, likely underestimates the actual changes of the synapses.

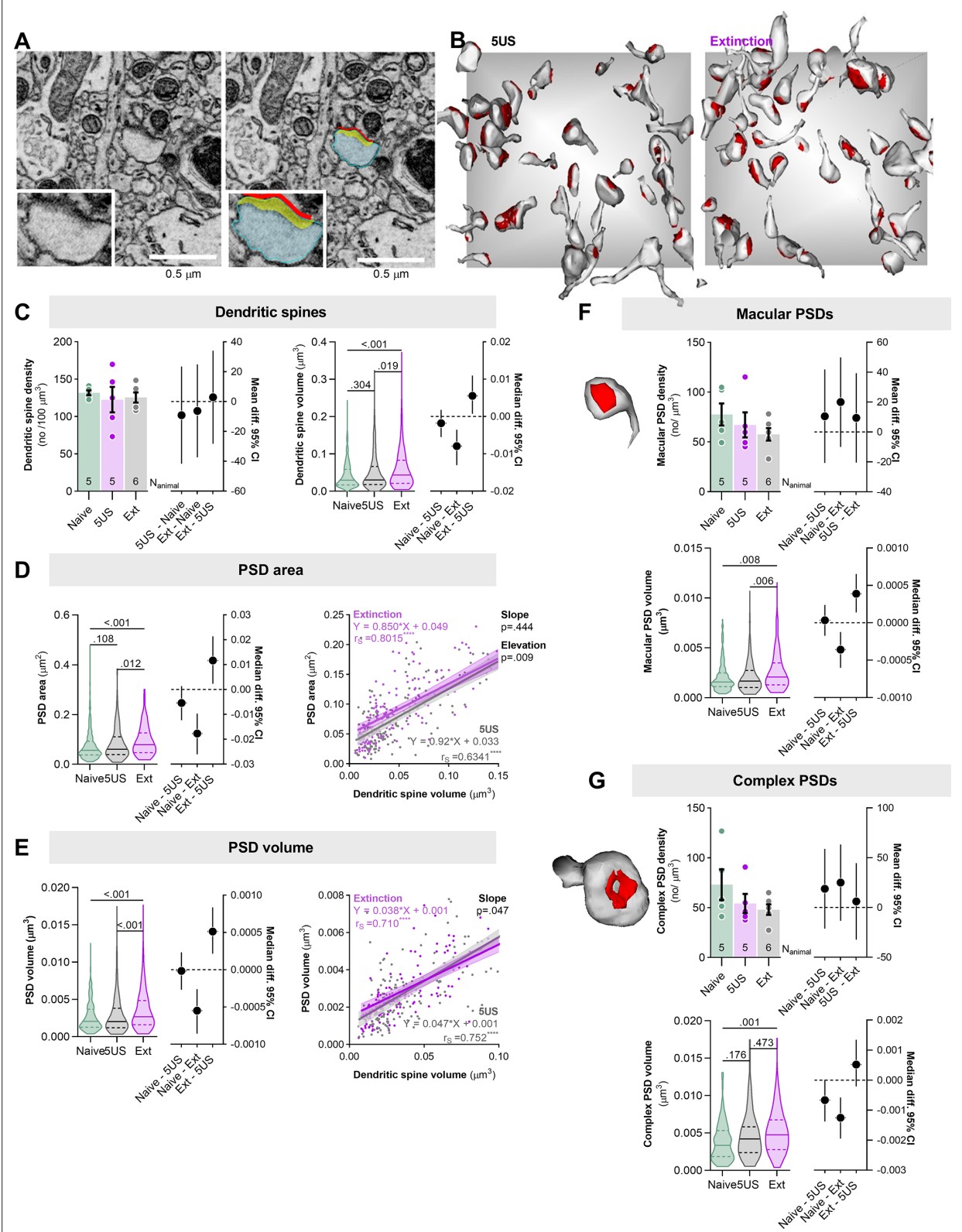

**Figure 5.** Contextual fear extinction remodels postsynaptic densities of the excitatory synapses in stratum lacunosum-moleculare (SLM). Mice underwent contextual fear conditioning (CFC) and were sacrificed 24 hr later (5US, n = 5) or after contextual fear extinction (CFE) session (Ext, n = 6). Naive animals were used as a control (n = 4). (**A, B**) The principles for serial block-face scanning electron microscopy (SBFSEM) analysis of dendritic spines and postsynaptic densities (PSDs) in SLM. (**A**) Tracing of a dendritic spine and PSD. A representative trace of a dendritic spine (blue), PSD surface

*Figure 5 continued on next page*

Figure 5 continued

area (red), and volume (yellow). (**B**) Exemplary reconstructions of dendritic spines and their PSDs from SBFSEM scans in SLM for the 5US and Ext groups. Dendritic spines and PSDs were reconstructed and analyzed in tissue bricks (6 × 6 × 2.5 μm). The gray background square is x = 6 × y = 6 μm. (**C–G**) Summary of SBFSEM data: (**C**) density of dendritic spines (F(2, 13)=0.2, p=0.829, means +/- SEM are shown; effect size for 5US-Naive: –9.08 [95CI 23.48; –41.64]; Ext-Naive: –6.19 [95CI 24.981; –37.377]; Ext-5US: 2.88 [95CI 34.06; –28.29]) and volume of dendritic spines (Kruskal–Wallis test with Dunn's post hoc tests, $U$ = 13.2, p=0.001, medians +/- IQR are shown; effect size for 5US-Naive: –0.0019 [95CI 0.0017; –0.0055]; Ext-Naive: –0079.19 [95CI –0.0035; –0.013]; Ext-5US: 0.0055 [95CI 0.010; 0.0007]; Naive N=315, 5US N=238, Ext N=259); (**D**) surface area of PSDs (Kruskal–Wallis test with Dunn's post hoc tests, $U$ = 17.9, p<0.001, medians +/- IQR are shown; effect size for 5US-Naive: –0.005 [95CI 0.0012; –0.012]; Ext-Naive: –0.017 [95CI 0.0096; –0.026]; Ext-5US: 0.011 [95CI 0.021; 0.0023]) and correlation of dendritic spine volume and PSD surface area (r$_s$, Spearman correlation r), each dot represents one dendritic spine; (**E**) volume of PSDs (Kruskal–Wallis test with Dunn's post hoc tests, $U$ = 17.2, p<0.001, medians +/- IQR are shown; effect size for 5US-Naive: –0.00001 [95CI 0.0002; –0.0002]; Ext-Naive: –0.0005 [95CI –0.0002; –0.0008]; Ext-5US: 0.0005 [95CI 0.0008; 0.0002]) and correlation of dendritic spine volume and PSD volume (r$_s$, Spearman correlation r), each dot represents one dendritic spine; (**F**) exemplary reconstruction of spine with macular PSD and data analysis for density of macular PSDs (one-way ANOVA, F(2, 13)=1.047, p=0.3788, means +/- SEM are shown; effect size for 5US-Naive: –10.57 [95CI 41.83; –20.679]; Ext-Naive: 20.04 [95CI 49.96; –9.88]; Ext-5US: –9.46 [95CI 39.389; –20.45]) and volume of macular PSDs (Kruskal–Wallis test with Dunn's post hoc tests, $U$ = 12.2, p=0.002, medians +/- IQR are shown; effect size for 5US-Naive: 0.00003 [95CI 0.0002; –0.00017]; Ext-Naive: –0.0003 [95CI –0.0001; –0.0006]; Ext-5US: 0.0003 [95CI 0.0006; 0.0001]; Naive N=212, 5US N=163, Ext N=168); (**G**) exemplary reconstruction of spine with complex PSD and data analysis for density of complex PSDs (one-way ANOVA, F(2, 13)=1.57, p=0.246, means +/- SEM are shown; effect size for 5US-Naive: 18.86 [95CI 58.9; –21.18]; Ext-Naive: 25.03 [95CI 63.37; –13.30]; Ext-5US: 6.17 [95CI 44.51; –31.16]) and volume of complex PSDs (Kruskal–Wallis test with Dunn's post hoc tests, $U$ = 12.6, p=0.002, medians +/- IQR are shown; effect size for 5US-Naive: –0.0007 [95CI 0.000012; –0.0013]; Ext-Naive: –0.001 [95CI –0.0006; –0.0019]; Ext-5US: 0.0005 [95CI 0.0013; –0.0002]; Naive N=103, 5US N=75, Ext N=91).

## Functional analysis of RE→dCA1 pathway during CFE

The structural analysis of RE axons and RE+ dendrites in dCA1 suggests that CFE strengthens RE→dCA1 synapses. To test whether the activity of RE→dCA1 pathway affects CFE, C57BL/6J mice were injected with AAV$_{2.1}$:DIO_hM4Di (hM4Di) or AAV$_{2.1}$:DIO_mCherry (control) into RE and with CAV2:Cre into dCA1 (*Figure 6A*), resulting in hM4Di or mCherry expression in the RE→dCA1 neurons (*Figure 6B*). Three weeks post-surgery and viral infection, mice underwent CFC, followed by CFE session and CFE test (*Figure 6C*). Mice were injected with CNO (i.p., 3 mg/ kg) or saline 30 min prior to the Extinction. The freezing levels increased within the conditioning session (pre- vs. post-US) and did not differ between the drug groups both in hM4Di and control animals. During the Extinction session, the mice from two drug groups had similar freezing levels at the beginning of the session and decreased freezing during the session, indicating no impairment of fear memory recall and successful within-session CFE. CFE memory was tested 24 hr after the Extinction session in the same context (Test). The hM4Di mice from the CNO group had higher levels of freezing compared to the saline animals, suggesting impaired long-term CFE memory (*Figure 6D*). We did not observe such an effect of CNO in the control group (*Figure 6E*).

As RE was linked with fear generalization (*Xu and Südhof, 2013*), we also tested the role of RE→dCA1 pathway in the recognition of the training context. In addition, as within-session CFE was not impaired in our first experiment, we tested whether the RE→dCA1 pathway also participates in the formation of short-term CFE memory. Mice underwent CFC, followed by a CFE session (Extinction 1). Next, we tested short-term (Test 1, 2 hr post Extinction 1) and long-term CFE memory (Extinction 2, 24 hr post Extinction 1) as well as fear generalization in the context B (CtxB, 2 hr post Test 2). Mice were injected with CNO or saline 30 min prior to the Extinction 1 and Test 2 (*Figure 6F*). As for the first cohort, we did not observe any differences in the freezing levels between saline and CNO groups during the CFC and Extinction 1. Mice in both groups successfully formed contextual fear memory, as indicated by the high levels of freezing in the training context at the beginning of the Extinction 1, and decreased freezing level within this session. However, the CNO group, compared to the saline animals, showed higher freezing levels both during a short-term (Test 1) and long-term CFE memory test (Extinction 2). They showed, however, no difference in freezing levels in the CtxB (*Figure 6C*). Hence, our results indicate that RE→dCA1 regulates formation of short- and long-term CFE memory but not recognition of the training context.

Since RE→dCA1 neurons bifurcate and innervate also other brain areas (*Hoover and Vertes, 2012*), in the following experiment, we tested whether specific inhibition of RE→dCA1 axons affects CFE. Mice were stereotaxically injected into RE with AAV$_{2.1}$:hSyn_hM4Di _mCherry, or AAV$_{2.1}$:camk2a_ mCherry (control) as a control (*Figure 6D–F*). Next, we implanted cannulas into dCA1 to deliver saline or CNO locally. Twenty-one days after the surgery and virus expression, mice underwent CFC,

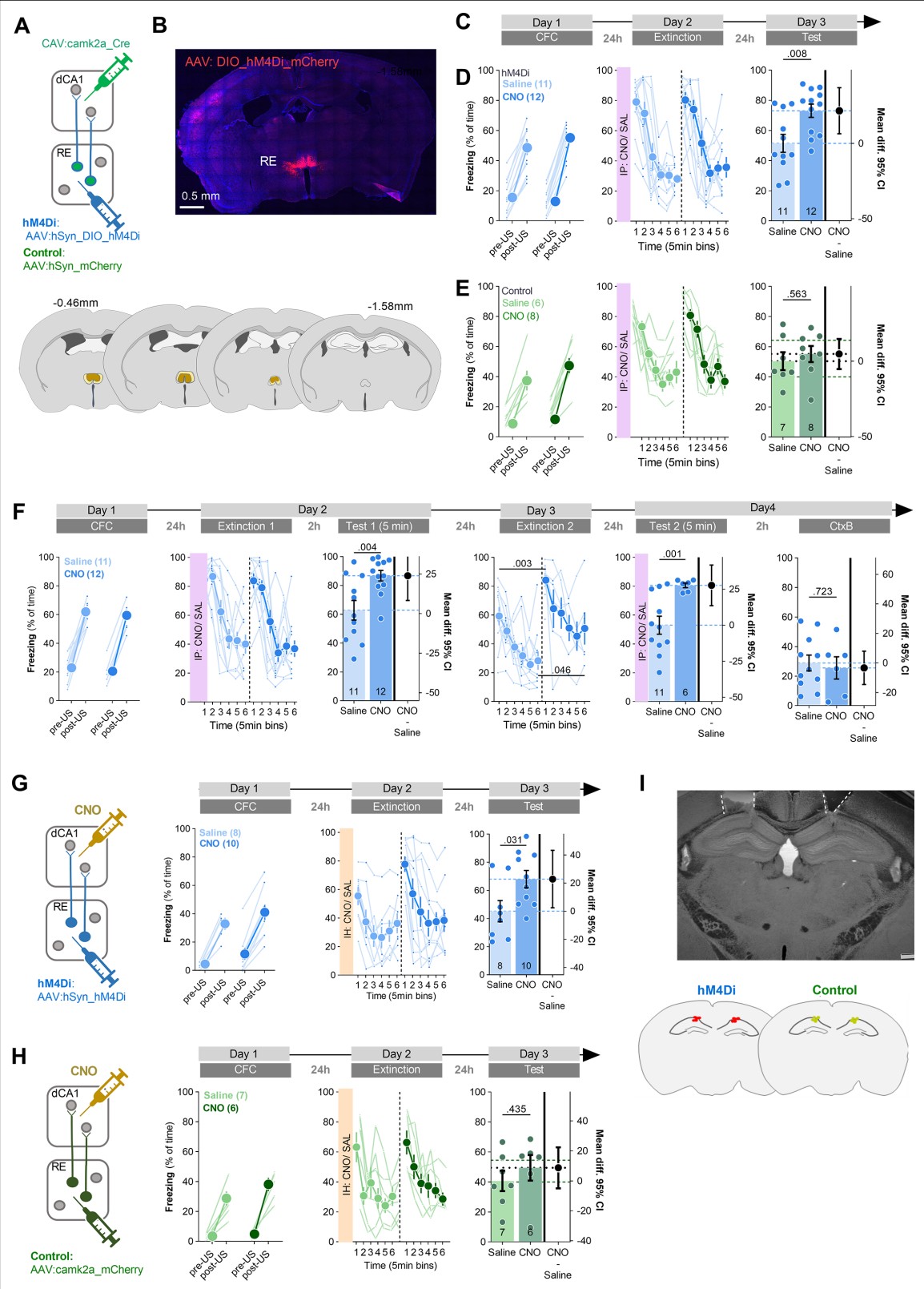

**Figure 6.** Chemogenetic inhibition of RE→dCA1 impairs extinction of contextual fear memory. (**A**) Experimental design. Mice received stereotaxic injection of AAV$_{2.1}$:hSyn_DIO_hM4Di_mCherry (hM4Di) or control virus (AAV$_{2.1}$:hSyn_mCherry) into nucleus reuniens (RE), and CAV$_2$:Cre_GFP into dCA1. (**B**) Representative microphotographs of hM4Di expression in RE and the extent of viral expression (blue, minimal; dark blue, maximal). (**C**) Experimental timeline. (**D**) Summary of data for freezing levels of the hM4Di group during the training (Training: RM ANOVA with Fisher's LSD post hoc test, effect of

*Figure 6 continued on next page*

*Figure 6 continued*

time: F(1, 21)=446, p<0.001; effect of CNO: F(1, 21)=0.165, p=0.689; time × CNO interaction: F(1, 21)=6.64, p=0.018), fear extinction session (Extinction: RM ANOVA with Šidak's post hoc test, effect of time: F(1, 21)=95.55, p<0.001; effect of CNO: F(1, 21)=0.259, p=0.616; time × CNO interaction: F(1, 21)=0.1740, p=0.681), and fear extinction test (Test: unpaired *t*-test: *t*(20) = 3.26, p=0.002; effect size: 25.442 [95CI 9.492; 41.392]). (**E**) Summary of data for freezing levels of the control group during the training (Training: RM ANOVA with Fisher's LSD post hoc test, effect of time: F(1, 10)=119 p<,001; effect of CNO: F(1, 10)=2,86 p=,122; time × CNO interaction: F(1, 10)=3,13 p=,107), fear extinction session (Extinction: RM ANOVA with Šidak's post hoc test, effect of time: F(2,897, 31,86)=21,32, p<0.001; effect of CNO: F(1, 11)=2,320,, p=0.156; time × CNO interaction: F(5, 55)=1,472, p=0.240) and fear extinction test (Test: unpaired *t*-test: *t*(10) = 0.149, p=0.884; effect size: 1.362 [95CI –19.016; 21.749]). (**F**) Experimental timeline and summary of data for freezing levels during the training. After stereotactic injection of DIO_hM4Di into RE and CAV:Cre_GFP into dCA1, mice underwent CFC (Training: RM ANOVA with Šidak's post hoc test, effect of time: F(1, 14)=218, p<0.001; effect of CNO: F(1, 14)=0.001, p=0.974; time × CNO interaction: F(1, 14)=0.010, p=0.919), CFE session (Extinction 1: RM ANOVA with Šidak's *post hoc* test, effect of time: F(1, 14)=45.5, p<0.001; effect of CNO: F(1, 14)=0.121, p=0.73; time × CNO interaction: F(1, 14)=0.020, p=0.89), short-term CFE memory test (Test 1: *t*-test: *t*(14) = 2.45, p=0.028; effect size: 23.935 [95CI 8.310; 39.561]), long-term CFE memory test (Extinction 2: RM ANOVA with Šidak's post hoc test, effect of time: F(1, 14)=23.1, p<0.001; effect of CNO: F(1, 14)=10.5, p=0.006; time × CNO interaction: F(1, 14)=0.028, p=0.87) and exposure to the context B (CtxB: unpaired *t*-test: *t*(14) = 0.057, p=0.955; effect size: –3.306 [95CI –22.850; 16.239]). Due to technical problems, only 6 mice out of 12 from the CNO group were trained on day 4 (Test 2 and CtxB). (**G**) Experimental timeline and summary of data for freezing levels during the training. Mice were stereotaxically injected into RE with AAV$_{2.1}$ encoding hM4Di and cannulae placed into dCA1. Next, mice underwent CFC (Training: RM ANOVA with Šidak's post hoc tests, effect of time: F(1, 12)=146, p<0.001; effect of CNO: F(1, 12)=1.60. p=0.23; time × CNO interaction: F(1, 12)=0.12, p=0.72), CFE session (Extinction: RM ANOVA with Šidak's post hoc test, effect of time: F(1, 15)=36, p<0.001; effect of CNO: F(1, 15)=2.14, p=0.16; time × CNO interaction: F(1, 15)=4.59, p=0.49) and long-term fear extinction memory test (unpaired *t*-test, *t*(15) = 2.30. p=0.036; effect size: 22.803 [95CI 2.383; 43.223]). (**H**) Experimental timeline and summary of data for freezing levels during the training. Mice were stereotaxically injected into RE with AAV2.1 encoding mCherry and cannulae were placed into dCA1. After surgery, they underwent CFC (Training: RM ANOVA with Šidak's post hoc tests, effect of time: F(1, 10)=84.3, p<0.001; effect of CNO: F(1, 10)=2.18, p=0.17; time × CNO interaction: F(1, 10)=1.61, p=0.23), CFE session (RM ANOVA with Šidak's post hoc test, effect of time: F(1, 11)=25.8, p<0.001; effect of CNO: F(1, 11)=0.006, p=0.93; time × CNO interaction: F(1, 11)=0.12, p=0.72) and CFE memory test (unpaired *t*-test, *t*(11) = 1.12, p=0.28; effect size: 13.718 [95CI 13.256; 40.696]). (**I**) Microphotography and schematic representations of cannulae placement. Means +/- SEM are shown.

followed by a CFE session and Test (***Figure 6D–F***). During the training, mice had low levels of freezing before US (pre-US) (***Figure 6D***). Freezing levels increased across a CFC session (post-US) and did not differ between the drug groups. Twenty-four hours later, mice received bilateral intra-hippocampal injection of CNO (3 µM) or saline and were re-exposed to the training context (Extinction). In the hM4Di virus group, mice injected with CNO showed higher freezing during the first 5 min of Extinction compared to the mice injected with saline. During Extinction, the freezing level decreased and did not differ between the groups at the end of the session. However, when CFE memory was tested (Test), the hM4Di mice injected with CNO had higher levels of freezing compared to the mice injected with saline. Similarly, as in our former study (***Tomaszewski et al., 2024***), freezing levels in the control CNO and saline groups differed neither during the CFC, nor during Extinction and Test (***Figure 6F***).

Overall, these results indicate that transient inhibition of RE→dCA1 has no effect on working memory (changes of freezing behavior within CFE session) but impairs both short- and long-term CFE memory even after two CFE sessions. We have also observed that it has no effect on contextual fear memory generalization but enhances contextual fear recall in mice that had CNO delivered to dCA1 through cannulas. Interestingly, in the saline groups with cannulas, the freezing levels were lower compared to the control groups in the experiments without cannulas (58–62% vs. 68–83%) (***Figure 6A and C***), suggesting that cannulas implantation impaired memory formation, possibly due to damage to the cortex (***Figure 6E***). Hence, inhibition of RE→dCA1 may enhance contextual fear recall only when the memory trace is suboptimal (***Figure 6G***), while CFE is regulated by this pathway across many experimental conditions (***Figure 6C, F and G***).

## The impact of RE on CFE-induced PSD-95 expression in dCA1

Our data indicate that CFE induces structural plasticity of the excitatory synapses and PSD-95 expression (an indicator of molecular remodeling of excitatory synapses) on RE+ dendrites in SLM. To test whether RE affects CFE-induced PSD-95 expression in dCA1, C57BL/6J mice were injected with hM4Di or control virus into RE (***Figure 7A–C***). Three weeks post-surgery and viral infection, mice underwent CFC, followed by a CFE session. Mice were injected with CNO (i.p., 3 mg/ kg) or saline 30 min prior to the Extinction (***Figure 7A***). The freezing levels increased within the conditioning session (pre- vs. post-US) and did not differ between the experimental groups. During the Extinction session, the mice from four experimental groups had similar freezing levels at the beginning of the session and decreased freezing during the session, indicating no impairment of fear memory recall and successful

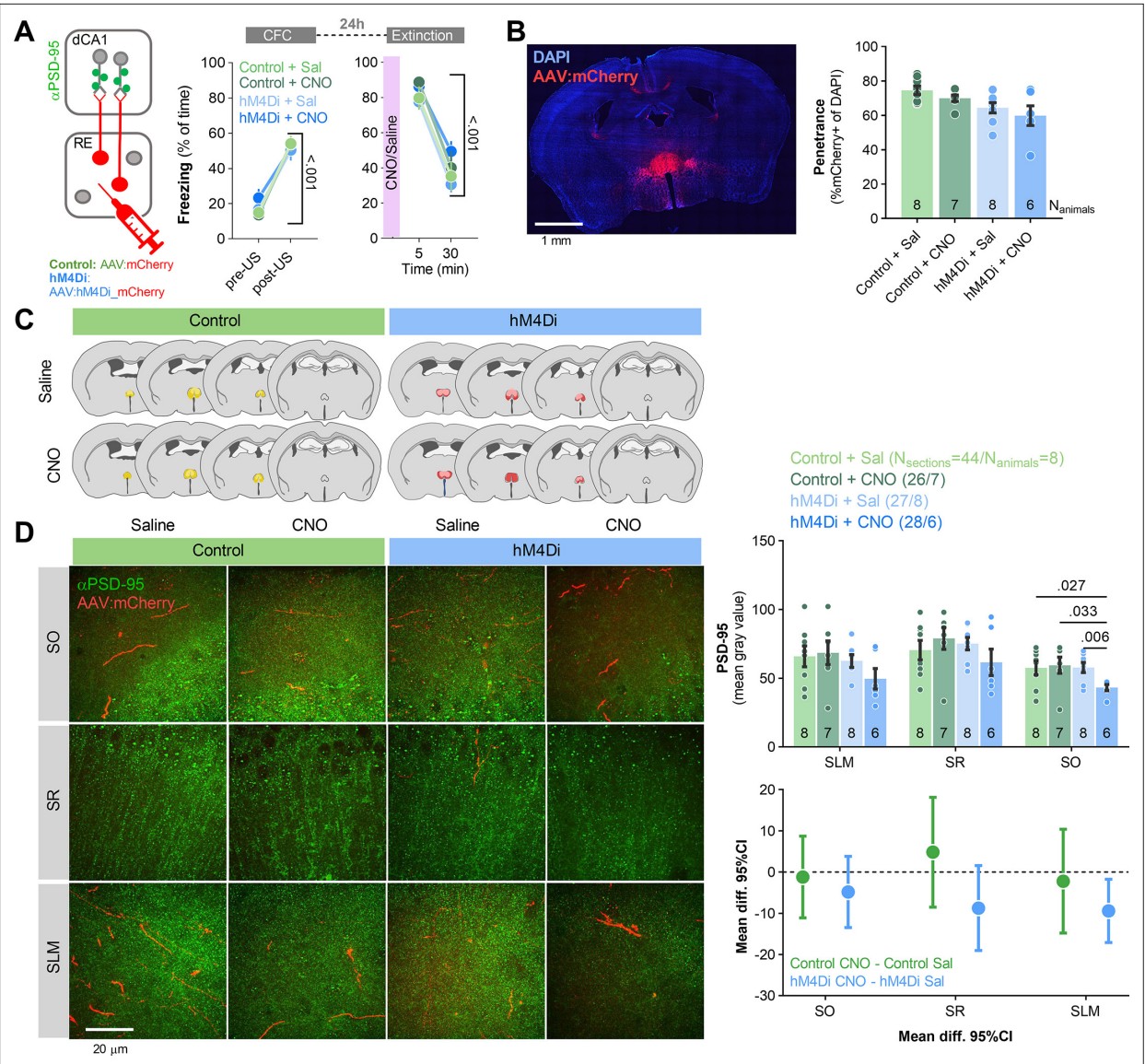

**Figure 7.** Nucleus reuniens (RE) regulates PSD-95 expression in dCA1 during contextual fear extinction. (**A**) Experimental timeline and summary of data showing freezing levels during contextual fear conditioning (CFC) and contextual fear extinction (CFE). After injection of $AAV_{2.1}$ encoding mCherry (control) or hM4Di into RE, mice underwent CFC (two-way ANOVA, effect of training, $F(1, 26)=322$, $p<0.001$; effect of group, $F(3, 26)=0.094$, $p=0.962$) and fear extinction session (effect of training, $F(1, 26)=144$, $p<0.001$; effect of group, $F(3, 26)=1.12$, $p=0.358$). Mice were injected with CNO or saline 30 min before Extinction and were sacrificed immediately after the session. The dCA1 brain slices with visible RE axons were used for PSD-95 immunostaining. (**B, C**) AAVs expression analysis in RE. (Left) Confocal scan of RE with local expression of mCherry. (Right) Penetrance of mCherry in experimental groups. (**C**) The extent of viral expression. (**D**) Representative images of PSD-95 immunofluorescent staining (green) and RE axons (red) in three strata of dCA1 (left) and data analysis (three-way RM ANOVA with post hoc LSD test for planned comparisons, effect of virus: $F(1, 21)=8.58$, $p=0.008$; effect of CNO: $F(1, 21)=0.566$, $p=0.460$; virus × CNO interaction: $F(1, 21)=7.72$, $p=0.011$; effect size for control CNO – control sal, SO: –1.190 [95CI 8.703; –11.085]; SR: 4.557 [95CI 18.149; –8.494]; SLM: –2.197 [95CI 10.410; –14.806]; effect size for hM4Di CNO – hM4Di sal, SO: –4.891 [95CI 3.427; –13.411]; SR: –8.757 [95CI 1.587; –19.697]; SLM: –9.481 [95CI –1.725; –17.037]). Means +/- SEM are shown.

within-session fear extinction. Mice were sacrificed immediately after Extinction and their brains were sliced. We analyzed the changes of PSD-95 protein levels in three dCA1 strata using fluorescent immunostaining.

PSD-95 levels were significantly lower in SLM in the hM4Di group treated with CNO compared to the control group treated with CNO and the hM4Di group treated with saline (*Figure 7D*). Such differences were not observed in other strata. Hence, chemogenetic inhibition of RE affected CFE-induced molecular changes of the excitatory synapses within RE input areas, possibly affecting the integration

of information from other regions involved in CFE (*Baldi and Bucherelli, 2014*; *Baldi and Bucherelli, 2015*; *Bevilaqua et al., 2006*; *Ji and Maren, 2008*; *Cai et al., 2018*).

## Discussion

RE, dCA1, and RE→CA1 projections are necessary for the formation and retrieval of fear extinction memories (*Venkataraman and Dias, 2023*; *Cassel et al., 2013*; *Totty et al., 2023*; *Ratigan et al., 2023*). However, the cellular and molecular mechanisms in the RE→CA1 projections that contributed to CFE remained unknown. Here, we demonstrate that RE axons are present in SO and SLM of dCA1 and chemogenetic inhibition of RE impairs excitatory synaptic transmission in both striata. Moreover, we discovered that in the SLM CFE upregulates the levels of synaptic scaffold protein, PSD-95 (*Chen et al., 2011*; *El-Husseini et al., 2000*; *Kornau et al., 1995*), accompanied by structural plasticity of RE→dCA1 synapses—growth of RE boutons and dendritic spines located on the RE+ dendrites. Finally, chemogenetic inhibition of the RE→dCA1 pathway impaired CFE, while chemogenetic inhibition of RE blocked the CFE-induced expression of PSD-95 in SLM. In summary, our observations support a framework in which the RE→SLM pathway participates in the updating of fearful context value by actively regulating CFE-induced molecular and structural synaptic plasticity in the SLM.

RE projects to the hippocampal formation and forms asymmetric synapses with dendritic spines and dendrites, suggesting innervation of excitatory synapses on both excitatory and aspiny inhibitory neurons (*Wouterlood et al., 1990*; *Herkenham, 1978*; *Kajiwara et al., 2008*; *Vertes et al., 2006*). This conclusion was confirmed by the electrophysiological studies showing that RE modulates function of the pyramidal cells by monosynaptic excitatory and polysynaptic inhibitory mechanisms (*Dolleman-Van der Weel et al., 1997*; *Kajiwara et al., 2008*). Here, we observed RE boutons in close apposition to dCA1 dendritic spines containing PSD-95-positive puncta, confirming innervation of the pyramidal neurons via excitatory synapses. The RE axons were found both in SO and SLM of dCA1, with the latter having much bigger boutons (*Figure 1A–G*). Furthermore, we recorded field excitatory synaptic potentials (fEPSPs) in dCA1 while chemogenetically inhibiting RE axons. This manipulation decreased fEPSPs both in SO and SLM, and had no effect on this parameter in SR (*Figure 1J and K*). Hence, our data supports the notion that RE activity increases excitatory synaptic transmission both in SO and SLM. Since RE projections were described so far only in SLM, the observation of RE axons in SO is surprising. This discrepancy with former studies may stem from the fact that RE axons in SO are thinner and have much smaller boutons compared to SLM. Hence, RE axons in SO are clearly visible only with confocal imaging of high magnification (×100, see *Figure 1C vs. D*). This fact could have been overlooked in the former studies that either used light microscopy or show confocal microphotographs taken with low magnification. Hence, in the future, it will be important to clarify the functional differences between RE→SO and RE→SLM pathways.

Both formation (*Abraham et al., 2019*; *Bliss and Collingridge, 1993*; *Morris et al., 2003*; *Radwanska et al., 2011*) and extinction of contextual fear memories (*Ziółkowska et al., 2023*; *Schuette et al., 2020*; *Stansley et al., 2018*; *Garín-Aguilar et al., 2012*) involve structural synaptic plasticity in dCA1. While most of the former studies focused on training-induced synaptic changes at more remote time points (>2 hr after training), some experiments have demonstrated that behavioral training may result in rapid (<45 min) synapse remodeling, synaptogenesis, and synapse elimination (*Ziółkowska et al., 2023*; *Gipson et al., 2013a*; *Gipson et al., 2013b*). In particular, we have previously demonstrated that 30-min-long CFE session induced structural synaptic plasticity in dCA1 pyramidal neurons in a stratum-specific manner, and it was accompanied by phosphorylation of PSD-95 at serine 73 and alterations of PSD-95 protein levels (*Ziółkowska et al., 2023*). We also showed that phosphorylation of PSD-95 at serine 73 in dCA1 is required for CFE and CFE-induced structural plasticity of SO synapses. Thus, rapid and PSD-95-dependent structural synaptic plasticity is an important step in the regulation of the dCA1 circuit during CFE. Here, we observed that CFE-induced increase of PSD-95 levels per dendritic spine and morphological changes of dendritic spines occurred in SO and SLM, but not SR, while the changes of dendritic spine density were observed only in SO. Hence, we confirmed the earlier findings. The only minor discrepancy with our former work is the observation of slightly bigger dendritic spines in SLM in the Ext vs. 5US group that we report here. As we assumed that this discrepancy could be a result of small differences between the groups, the analysis was repeated using 3D SBFSEM. We confirmed that the mice in the Ext group had larger dendritic spines in SLM compared to the 5US group. Interestingly, they also had much larger PSDs. Hence, our data

indicate the importance of the analysis of structural synaptic plasticity using 3D SBFSEM and indicate that the confocal analysis of dendritic spine volume as a proxy of synaptic changes may underestimate the actual changes of the synapses. So far, coupling of PSD size to spine volume was assumed to be a permanent spine feature (*Arellano et al., 2007*). Uncoupling of PSD and spine volumes was shown to be transient (<7 min) after glutamate uncaging and caused by slower growth of a PSD compared with a spine (*Bosch et al., 2014*; *Meyer et al., 2014*). Here, similarly as in our former studies (*Ziółkowska et al., 2023*; *Borczyk et al., 2019*; *Śliwińska et al., 2020*), we show that PSD-core growth exceeds dendritic spine growth for at least 30 min. The precise reasons for the correlation between dendritic spine volume and PSD-core volume, and their coupling during synaptic plasticity, remain unknown. Changes in the ratio between these two compartments likely impact the stoichiometry between proteins in the PSD and those in the dendritic spine, influencing the properties of their complexes, long-term synaptic stability, synaptic strength, and dendritic spine sizes.

In addition, the current study specifies that CFE-induced structural and molecular changes of dendritic spines occur on RE+ dendrites in SLM, and RE- dendrites in SO. In particular, CFE results in spinogenesis of RE+/+ spines and increased median size of RE boutons, RE+/+ and RE+/- spines in SLM. Thus, while former studies linked the behavioral functions of RE with its impact on the coherence of theta oscillations between the hippocampus and prefrontal cortex (*Griffin, 2021*; *Totty et al., 2023*; *Ito et al., 2015*), we extend these findings demonstrating the impact of RE on training-induced structural and molecular plasticity of the excitatory synapses in dCA1 that participates in CFE.

Although RE+/+ dendritic spines are sparse in dCA1 (based on mCherry AAV penetrance in RE and frequency of RE+/+ dendritic spines in our study, we asses that they constitute 5–6% of all synapses), our findings show that RE has a very significant impact on dCA1 excitatory synapses. In particular, we have observed that chemogenetic inhibition of RE prevents CFE-induced expression of PSD-95 globally in SLM, suggesting the effect of RE on dCA1 synapses that spreads far beyond RE+/+ inputs. Moreover, we observed differential CFE-induced structural synaptic plasticity on RE+/- and RE-/- spines, suggesting heterosynaptic contribution of RE to plasticity of RE+/- dendritic spines. In agreement with these findings, *Dolleman-Van der Weel et al., 1997* demonstrated that RE stimulation produced large negative-going field potentials at SLM, indicative of prominent depolarizing actions on distal apical dendrites of CA1 pyramidal cells. Hence, RE may exert a persistent influence on the state of pyramidal cell excitability, depolarizing cells to close to threshold for activation by other excitatory inputs. Consistent with this, *Bertram and Zhang, 1999* compared the effects of stimulation of the RE with stimulation of the CA3 region of the hippocampus on population responses at CA1 and reported that RE actions on CA1 were in some cases considerably greater than those of CA3 on the CA1. Moreover, intra-RE infusions of ketamine and muscimol bidirectionally regulate neuronal firing and delta power in CA1 (*Zhang and Lisman, 2012*; *Duan et al., 2015*; *Roy et al., 2017*). Altogether, this data indicates that RE may impact general activity of CA1 neurons. Although the mechanism is poorly understood, the regulation of synaptic protein expression, including PSD-95, beyond RE+/+ synapses is likely important. In addition, the effects of RE on synaptic levels of PSD-95 may affect the integration of information from other dCA1 inputs that regulate CFE, such as the entorhinal cortex or CA3 area (*Baldi and Bucherelli, 2014*; *Baldi and Bucherelli, 2015*; *Bevilaqua et al., 2006*; *Ji and Maren, 2008*; *Cai et al., 2018*).

The former studies demonstrated the role of the RE in fear memory encoding, retrieval, extinction, and generalization (*Ramanathan et al., 2018*; *Xu and Südhof, 2013*; *Troyner and Bertoglio, 2021*; *Ramanathan and Maren, 2019*; *Vasudevan et al., 2022*; *Vetere et al., 2017*). Furthermore, recent work has shown the role of the RE→dCA1 pathway in extinction of contextual fear and recall of CFE (*Totty et al., 2023*; *Ratigan et al., 2023*). In agreement with these findings, inhibition of the RE→dCA1 neurons and RE axons in dCA1 blocked the formation of fear extinction memory in our experiments. In addition, we observed that transient chemogenetic inhibition of RE→dCA1 during the first fear extinction session resulted in impaired extinction not only during the short- and long-term CFE memory test but also throughout the entire second extinction session. This persistent impairment of CFE alludes to long-term extinction impairment induced by PSD-95 mutation in dCA1 observed in our former study. We found that contextual fear memory cannot be updated by the animals with dCA1 phosphorylation-deficient PSD-95(S73A) mutation even if they undergo six 30 min CFE sessions (*Ziółkowska et al., 2023*). Hence, our data support the framework in which CFE-induced and PSD-95-dependent synaptic plasticity in dCA1 is necessary to reconsolidate and link the memories of

contextual fear and CFE in order to persistently suppress contextual fear. The need for interaction between these two memory traces for fear extinction to occur, and the contribution of RE in this process, was also suggested by *Totty et al., 2023*. The RE-regulated synaptic alterations of PSD-95 levels that spread far beyond RE+/+ synapses may facilitate integration of contextual fear and CFE information that are likely generated by different dCA1 inputs (*Baldi and Bucherelli, 2014*; *Baldi and Bucherelli, 2015*; *Bevilaqua et al., 2006*; *Ji and Maren, 2008*; *Cai et al., 2018*). Thus, we assume that the integration of all inputs to dCA1, conveying different aspects of the cognitive process (*Baldi and Bucherelli, 2015*; *Bevilaqua et al., 2006*), is required for optimal extinction of contextual fear.

Previous work has found that RE is necessary for proper discrimination of contexts (*Xu and Südhof, 2013*), and reductions in context discrimination might result in a relapse of fear (*Bouton and Bolles, 1979*). However, more recent studies showed that the activity of RE axons in CA1 is not contextually modulated (*Ratigan et al., 2023*) and that 8 Hz stimulation of RE does not impair context recognition, although it affects recall of CFE (*Totty et al., 2023*). We also see neither significant changes of RE+/+ synapses upon exposure to neutral context nor impairments of context recognition after chemogenetic inhibition of RE→CA1. Hence, our work supports the notion that impaired fear extinction upon RE inactivation is a result of deficits in memory integration, rather than impaired recognition of the extinction context.

Overall, our study illustrates a novel synaptic mechanism that participates in updating of contextual fear memories. This mechanism involves structural and molecular synaptic plasticity of the RE→dCA1 projections as well as global changes of PSD-95 levels in dCA1 area. Since memories may be repeatedly reorganized upon recall (*Nader et al., 2000*; *Schafe et al., 2001*), the molecular and cellular mechanisms involved in extinction of the existing fearful, aversive, or appetitive memories provide excellent targets for memory impairment therapies. In particular, understanding the mechanisms that underlie CFE may be relevant for post-traumatic stress disorder treatment.

## Materials and methods

### Animals

The Thy1-GFP(M) (The Jackson Laboratory, JAX:007788, RRID:IMSR_JAX:007788) mutant mice were bred as heterozygotes at Nencki Institute and PCR genotyped (The Jackson Laboratory protocol). We used both males and females in age- and sex-balanced groups. C57BL/6J mice were purchased from Białystok University, Poland. Mice were 3–4 months old at the time of training. The mice were housed alone and maintained on a 12 hr light/dark cycle with food and water ad libitum. All experiments were undertaken in accordance with the Poland Animals (Scientific Procedures) Act and approved by the Local Ethics Committee in Warsaw, Poland (no. 529/2018).

### Stereotactic surgery

Mice were anesthetized with isoflurane (5% for induction, 1.5–2.0% after) and fixed in the stereotactic frame (51503, Stoelting, Wood Dale, IL). $AAV_{2.1}$ viruses (camk2a_mCherry [titer 7.5 * $10^8$/µl, from Karl Deisseroth's Lab]; hSyn_hM4Di_mCherry [4.59 * $10^9$/µl; Addgene plasmid #50475]; hSyn_DiO_hM4Di_mCherry [4.2 * $10^{10}$/µl; Addgene plasmid #44362]) were injected into RE (150 nl, coordinates from the Bregma: AP, –0.58 mm; ML, 0 mm; DV, 4.3 mm) and $CAV_2$ virus ($CAV_2$:Cre_GFP [13.1 * $10^9$/µl, PVM Montpellier]) was injected into the dCA1 region of the hippocampus (500 nl per site, coordinates from the Bregma: AP, –2.1 mm; ML,±1.1 mm; DV, –1.3 mm) (*Paxinos and Franklin, 2019*). We used beveled 26-gauge metal needle and 10 µl microsyringe (SGE010RNS, WPI, Sarasota, USA) connected to a microsyringe pump (UMP3, WPI), and its controller (Micro4, WPI) with an injection rate 0.1 µl/min. After injection, the needle was left in place for an additional 10 min to prevent unwanted spread of the vector. The AAV viruses were prepared by the Laboratory of Animal Models at Nencki Institute of Experimental Biology, Polish Academy of Sciences.

### Cannula placement

Mice were anesthetized by inhalation of 3–5% isoflurane (IsoFlo; Abbott Animal Health) in oxygen and positioned in a stereotaxic frame (51503, Stoelting). Two holes were drilled in the skull and a double guide cannulae (2 mm apart and 2 mm long; 26GA, Plastics One) was lowered into the holes such that the cannula tip was located over dorsal CA1 area (2 mm posterior to Bregma, ±1 mm lateral,

and −1.3 mm vertical). The cannulae were kept patent by using 33-gauge internal dummy cannulae (Plastics One). The animals were used in CFC 21 days after the cannulation. Animals received bilateral CNO (3 µM, 0.2 µl per side for 1 min; Tocris Bioscience, Cat# 4936) (*Stachniak et al., 2014*) or saline injections (0.2 µl per side) 30 min before Extinction session via intrahippocampal injection cannulae (33 gauge). After the infusion, the cannula was left in place for 30 s. The cannula placement was verified by histology, and only data from animals with correct cannula implants were included in statistical analyses.

## Contextual fear conditioning

Mice were trained in a conditioning chamber in a soundproof box (Med Associates Inc, St Albans, VT). The chamber floor had a stainless steel grid for shock delivery. Before training, chambers were cleaned with 70% ethanol and paper towels were soaked in ethanol and placed under the grid floor.

On the day of CFC, mice were brought to the room with a conditioning chamber 30 min before the training to acclimatize. Animals were placed in the chamber, and after a 148 s introductory period, a foot shock (2 s, 0.70 mA) (US) was presented. The shock was repeated five times, with intertrial interval of 90 s. Then, 30 s after the last shock, mice were returned to the home cage. Mice of the 5US group were sacrificed 24 hr after the initial training. The next day contextual fear was extinguished by re-exposing the animals to the conditioning chamber for 30 min without US presentation. Mice of the Extinction group were sacrificed right after the Extinction session. To test the efficiency of extinction, mice were re-exposed to the training context 24 hr after the extinction session for an additional 5 min (Test).

A video camera was fixed inside the door of the sound attenuating box for the behavior to be recorded and automatically scored with VideoFreeze Software (Med Associates Inc). For the assessment of learning, we used percent of time spent by animals freezing (% freezing). Freezing behavior was defined as complete lack of movement, except respiration. To assess within-session learning (working memory), we compared pre- and post-US freezing frequency (the first 148 s vs. last 30 s) during the CFC (day 1). To assess the formation of long-term contextual fear memory, we compared pre-US freezing (day 1) and the first 5 min of the Extinction session (day 2). To assess within-session CFE, we ran two-way ANOVA to assess the effect of time and manipulation on freezing frequency. Freezing data were analyzed in 5 min bins. To assess the formation of long-term CFE memory, we compared the first 5 min of the Extinction session (day 2) and Test session (day 3).

## CNO administration

Clozapine N-oxide (CNO) (Tocris Bioscience, Cat# 4936) was dissolved in 0.9% saline. 3 mg/kg CNO was injected 30 min prior to the CFE session.

## Immunostaining

Mice were anesthetized and perfused with 4% PFA (Sigma-Aldrich) in PBS buffer, pH 7.5 (Medicago). Coronal brain sections (40 µm) were prepared (Cryostats Leica CM1950) and stored at −20°C in PBSAF (PBS, 15% sucrose [POCH], 30% ethylene glycol [Sigma-Aldrich], and 0.05% NaN₃ [Sigma-Aldrich]). The sections were washed three times in a PBS buffer and incubated in a blocking solution (5% NDS [Jackson Immunoresearch]/0.3% Triton X-100 [Sigma-Aldrich] in PBS) for 1 hr at room temperature (RT). Next, sections were incubated at 4°C overnight with primary antibodies directed against PSD-95 (1:500, Millipore, MAB 1598), washed three times in 0.3% Triton X-100 in PBS, and incubated at RT for 90 min with a secondary antibody bound with Alexa Fluor 647 (1:500, Invitrogen, A31571). The sections were washed three times in PBS, mounted on glass microscope slides (Thermo Fisher Scientific), and coverslipped with Fluoromount-G medium with DAPI for fluorescence (Invitrogen, 00-4959-52).

## Confocal microscopy and image quantification

The microphotographs of dendritic spines in the Thy1-GFP(M) mice, fluorescent PSD-95 immunostaining, and axons labeled with mCherry were taken on a Spinning Disc confocal microscope (63× oil objective, NA 1.4, pixel size 0.13 µm × 0.13 µm) (Zeiss, Göttingen, Germany). We took microphotographs (16 bit, z-stacks of 20–48 scans; 260 nm z-steps) of six secondary dendrites per region per animal from SO, SR, and SLM (in the middle of the strata) of dCA1 pyramidal neurons (AP, Bregma from

–1.7 to 2.06). The images were processed and analyzed semi-automatically as previously described (*Ziółkowska et al., 2023*). All image Z-stacks were first processed with Huygens Professional deconvolution software to reduce background and extract all relevant information from this image series in a statistically reliable way. The RE+ and RE- dendrites were identified by visual inspection of Z-stacks of images in three channels (mCherry [axons], GFP [dendrites], and αPSD-95 [excitatory synapse scaffolding]) and Colocalization Highlighter function (ImageJ 1.52n). Colocalizations were identified on individual confocal planes. RE+ dendrites were defined as dendrites with at least one RE→CA1 synapse (physical apposition of signals in all three channels and colocalizations identified with Colocalization Highlighter function [ImageJ 1.52n] in at least one confocal plane) within the analyzed Z-stack. Next, Z-stacks were transferred into maximal projections to analyze dendritic spines (area and linear density), PSD-95+ puncta (mean gray value of PSD-95+ signal per dendritic spine) and axonal boutons (linear density and area). All dendritic spines and axonal boutons were outlined manually and measured in ImageJ 1.52n software measure tool as previously described (*Ziółkowska et al., 2023*). Boutons were defined as any swelling of an en passant fiber that colocalized with at least one PSD-95+ puncta and with diameter at least three times bigger than axon. Dendritic spines were defined as any small protrusions of a dendrite. Length of RE axons, dendrites colocalizing with RE axon (dendrite RE+ with spine colocalizing or not with axons [RE+/+ and RE+/-]), and dendrites not colocalizing with RE axon (dendrite RE-) were analyzed within tissue bricks (67.7 × 67.7 × 5.4 µm; from z-stack of 20 microphotographs). All analyses were done by the experimenters blind to the experimental groups.

## Serial block-face scanning electron microscopy

Mice were transcardially perfused with cold phosphate buffer pH 7.4, followed by 0.5% EM-grade glutaraldehyde (G5882 Sigma-Aldrich) with 2% PFA in phosphate buffer pH 7.4 and postfixed overnight in the same solution. Brains were then taken out of the fixative and cut on a vibratome (Leica VT 1200) into 100 µm slices. Slices were kept in phosphate buffer pH 7.4, with 0.1% sodium azide at 4°C. Then, slices were washed three times in cold phosphate buffer and postfixed with a solution of 2% osmium tetroxide (#75632 Sigma-Aldrich) and 1.5% potassium ferrocyanide (P3289 Sigma-Aldrich) in 0.1 M phosphate buffer pH 7.4 for 60 min on ice. Next, samples were rinsed 5 × 3 min with double-distilled water (ddH$_2$O) and subsequently exposed to 1% aqueous thiocarbohydrazide (TCH) (#88535 Sigma) solution for 20 min. Samples were then washed 5 × 3 min with ddH$_2$O and stained with osmium tetroxide (1% osmium tetroxide in ddH$_2$O) for 30 min in RT. Afterward, slices were rinsed 5 × 3 min with ddH$_2$O and incubated in 1% aqueous solution of uranyl acetate overnight at 4°C. The next day, slices were rinsed 5 × 3 min with ddH$_2$O, incubated with lead aspartate solution for 30 min at 60°C, and then washed 5 × 3 min with ddH$_2$O and dehydration was performed using graded dilutions of ice-cold ethanol (30%, 50%, 70%, 80%, 90%, and 2 × 100% ethanol, 5 min each). Then slices were infiltrated with Durcupan resin and flat embedded between Aclar sheets (Ted Pella #10501-10). Next, Aclar layers were separated from the resin-embedded samples, and the dCA1 region was cut out with a razorblade. Squares of approximately 1 × 1 × 1 mm were attached to aluminum pins (Gatan metal rivets, Oxford Instruments) with very little amount of cyanacrylate glue. Next, samples were mounted to the ultramicrotome to cut 1-µm-thick slices. Slices were transferred on a microscope slide, briefly stained with 1% toluidine blue in 5% borate, and observed under a light microscope to confirm the region of interest. Next, samples were grounded with silver paint (Ted Pella, 16062-15) and left for drying for 4–12 hr, before the specimens were mounted into the 3View2 chamber.

## SBEM imaging and 3D reconstructions

Samples were imaged with Zeiss SigmaVP (Zeiss, Oberkochen, Germany) scanning electron microscope equipped with 3View2 GATAN chamber using a backscatter electron detector. Scans were taken in the middle portion of dCA1 SO. From each sample, 200 sections were collected (thickness 60 nm). Imaging settings: high vacuum with EHT 2.8 kV, aperture: 20 µm, pixel dwell time: 3 µs, pixel size: 5–6.2 nm. Scans were aligned using the ImageJ software (ImageJ → Plugins → Registration → StackReg) and saved as .tiff image sequence. Next, aligned scans were imported to Reconstruct software, available here (Synapse Web Reconstruct, RRID:SCR_002716). Dendritic spine density was analyzed from three bricks per animal with the unbiased brick method (*Fiala and Harris, 2001*) per tissue volume. Brick dimensions 3 × 3 × 3 µm were chosen to exceed the length of the largest profiles

in the data sets at least twice. To calculate the density of dendritic spines, the total volume of large tissue discontinuities was subtracted from the volume of the brick.

A structure was considered to be a dendritic spine when it was a definite protrusion from the dendrite, with electron-dense material (representing postsynaptic part of the synapse, PSD) on the part of the membrane that opposed an axonal bouton with at least three vesicles within a 50 nm distance from the cellular membrane facing the spine. For 3D reconstructions, PSDs and dendritic spines in one brick were reconstructed for each sample. PSDs were first reconstructed and second, their dendritic spines were outlined. To separate dendritic spine necks from the dendrites, a cutoff plane was used approximating where the dendritic surface would be without the dendritic spine. PSD volume was measured by outlining dark, electron-dense areas on each PSD-containing section (45). The PSD area was measured manually according to the Reconstruct manual. All non-synaptic protrusions were omitted in this analysis. For multisynaptic spines, the PSD areas and volumes were summed. All analyses were done by the experimenters blind to the experimental groups.

## Electrophysiology

Mice were deeply anesthetized with isoflurane, decapitated, and the brains were rapidly dissected and transferred into ice-cold cutting artificial cerebrospinal fluid (ACSF) consisting of (in mM) 87 NaCl, 2.5 KCl, 1.25 $NaH_2PO_4$, 25 $NaHCO_3$, 0.5 $CaCl_2$, 7 $MgSO_4$, 20 D-glucose, 75 saccharose equilibrated with carbogen (5% $CO_2$/95% $O_2$). The brain was cut to two hemispheres and 350-μm-thick coronal brain slices were cut in ice-cold cutting ACSF with Leica VT1000S vibratome. Slices were then incubated for 15 min in cutting ACSF at 32°C. Next, the slices were transferred to recording ACSF containing (in mM) 125 NaCl, 2.5 KCl, 1.25 $NaH_2PO_4$, 25 $NaHCO_3$, 2.5 $CaCl_2$, 1.5 $MgSO_4$, 20 D-glucose equilibrated with carbogen and incubated for minimum 1 hr at RT.

Extracellular field potential recordings were recorded in a submerged chamber perfused with recording ACSF in RT. The potentials were evoked with a Stimulus Isolator (A.M.P.I Isoflex) with a concentric bipolar electrode (FHC, CBARC75) placed in the SO, SR, or SLM of CA3 depending on the experiment. The stimulating pulses were delivered at 0.1 Hz, and the pulse duration was 0.3 ms. Recording electrodes (resistance 1–4 MΩ) were pulled from borosilicate glass (WPI, 1B120F-4) with a micropipette puller (Sutter Instruments, P-1000) and filled with recording ACSF. The recording electrodes were placed in the SO, SR, or SLM of the dorsal CA1 area depending on the measurement. Simultaneously, a second recording electrode was placed in the stratum pyramidale to measure population spikes. For each slice, the recordings were done in both SO and SR. fEPSPs were always recorded first in SR and then the stimulating and recording electrodes were moved to SO and SLM. Recordings were acquired with MultiClamp 700B (Molecular Devices, CA), Digidata 1550B (Molecular Devices), and Clampex 10.0 software (Molecular Devices). Input/output curves were obtained by increasing stimulation intensity by 25 μA in the range of 0–300 μA. Input/output curves were analyzed with AxoGraph 1.7.4 software (Axon Instruments, USA). The slope of fEPSP, relative amplitude of population spikes, and FV were measured.

## Statistics

Analysis was performed using GraphPad Prism 9. All the statistical details of experiments can be found in the legends of the figures. Sample size was determined based on published experiments and our experience in conducting similar experiments. Data with normal distribution are presented as mean ± SEM or as median ± IQR for the population with non-normal distribution. When the data met the assumptions of parametric statistical tests, results were analyzed by one- or repeated measures two-way ANOVA, followed by Tukey's or Fisher's post hoc tests, where applicable. Areas of axonal boutons, dendritic spines, and PSD-95 puncta did not follow normal distributions and were analyzed with the Kruskal–Wallis test. To facilitate the interpretation of our results, we followed the convention of defining $p < 0.05$ as significant in the text. Wherever possible, we used estimation-based statistics with mean-difference plots instead (*Ho et al., 2019*). The samples were excluded from the analysis only when evident technical problems occurred.

## Acknowledgements

This work was supported by the National Science Center (Poland) (PRELUDIUM grant no. 2016/23/N/NZ4/03304 to MZ; and MAESTRO grant no. 2020/38/A/NZ4/00483 to KR). The project was carried

out with the use of CePT infrastructure financed by the European Union—The European Regional Development Fund within the Operational Program 'Innovative economy' for 2007–2013. The funders had no role in study design, data collection and analysis, decision to publish, or preparation of the manuscript.

## Additional information

### Funding

| Funder | Grant reference number | Author |
|---|---|---|
| Narodowe Centrum Nauki | 2016/23/N/NZ4/03304 | Magdalena Ziółkowska |
| Narodowe Centrum Nauki | 2020/38/A/NZ4/00483 | Kasia Radwanska |

The funders had no role in study design, data collection and interpretation, or the decision to submit the work for publication.

### Author contributions

Magdalena Ziółkowska, Conceptualization, Data curation, Formal analysis, Funding acquisition, Investigation, Writing – original draft, Writing – review and editing; Narges Sotoudeh, Data curation, Writing – review and editing; Anna Cały, Monika Puchalska, Roberto Pagano, Malgorzata Alicja Śliwińska, Ahmad Salamian, Data curation, Formal analysis, Writing – review and editing; Kasia Radwanska, Conceptualization, Data curation, Formal analysis, Funding acquisition, Visualization, Writing – original draft, Project administration, Writing – review and editing

### Author ORCIDs

Anna Cały ⓘD https://orcid.org/0000-0002-7687-4471
Malgorzata Alicja Śliwińska ⓘD https://orcid.org/0000-0002-5016-4949
Kasia Radwanska ⓘD https://orcid.org/0000-0001-7445-6180

### Ethics

All experiments were undertaken in accordance with the Poland Animals (Scientific Procedures) Act and approved by the Local Ethics Committee in Warsaw, Poland (no. 529/2018).

Reviewer #1 (Public review): https://doi.org/10.7554/eLife.101736.3.sa1
Reviewer #2 (Public review): https://doi.org/10.7554/eLife.101736.3.sa2
Author response https://doi.org/10.7554/eLife.101736.3.sa3

## Additional files

### Supplementary files

Supplementary file 1. 95% CI for training-induced changes in density and area of dendritic spines in CA1. (a) 95% CI for training-induced changes in density of dendritic spines in SO. (b) 95% CI for training-induced changes in dendritic spine area in SO. (c) 95% CI for training-induced changes in PSD-95 protein levels in SO. (d) 95% CI for training-induced changes in density of dendritic spines in SR. (e) 95% CI for training-induced changes in dendritic spine area in SR. (f) 95% CI for training-induced changes in PSD-95 protein levels in SR. (g) 95% CI for training-induced changes in density of dendritic spines in SLM. (h) 95% CI for training-induced changes in dendritic spine area in SLM. (i) 95% CI for training-induced changes in PSD-95 protein levels in SLM.

MDAR checklist

### Data availability

Raw data and the code used for analysis of confocal data is available at OSF (https://osf.io/bnkpx/).

The following previously published dataset was used:

| Author(s) | Year | Dataset title | Dataset URL | Database and Identifier |
|-----------|------|---------------|-------------|-------------------------|
| Radwanska K | 2024 | RE-dCA1 projections in CFE | https://osf.io/bnkpx/ | Open Science Framework, bnkpx |

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
