## [Editor Report · eLife Assessment]

This work provides **important** findings characterizing potential synaptic mechanisms supporting the role of midline thalamus-hippocampal projections in fear memory extinction in mice. The methods and approaches were considered **solid**, though some evidence is incomplete as there are some concerns with the analytical approaches used for some aspects of the study. This work will be of interest to those in the field of thalamic regulation and fear memory.

---

## [Referee Report · Reviewer #1 (Public review)]

The findings of Ziolkowska and colleagues show that a specific projection from the nucleus reuniens of the thalamus (RE) to dorsal CA1 of the hippocampus plays an important role in fear extinction learning in male and female mice. In and of itself, this is not a new finding. Yet, the potential novelty and excitement comes from the authors' identification of structural alterations from RE projecting neurons to the specific stratum lacunosum moleculare subregion of CA1 after learning. The authors use a range of anatomical and functional approaches to demonstrate structural synaptic changes in dorsal CA1 that parallel the necessary role of RE inputs in modulating extinction learning. The significance of these findings was previously hampered by several technical shortcomings in the experimental design and interpretation. The authors adequately addressed some of the design concerns raised in the previous round, along with the interpretive critique that they couldn't localize the timing of effects to consolidation as originally claimed. Nevertheless, the authors provided an inadequate response to the concern regarding their misapplication of Ns and missing controls in one experiment.

In the previous review, a major methodological weakness in the experimental design involved the widespread misapplication of Ns used for the statistical analyses. Much of the anatomical analyses of structural synaptic changes in the RE-CA1 pathway used N = number of axons (Figs. 1, 2), N = number of dendrites (Figs. 3, 4), and N = number of sections (Fig. 7). In each instance it was recommended that N = animal number should be used. Reasons for this are as follows: this is standard practice in neuroanatomical research; using N = branch/ dendrite/ bouton/ spine number artificially inflates the statistical power and this incorrectly assumes independence of observations; using N = number of sections, etc., doesn't account for imbalances in the number of observations that vary from animal to animal that may skew group results.

In the authors' response, they generally concurred, but then they followed up with the defense that the number of items was too few in some cases, or absent in others, to permit using N = animal number. While they changed some of their data to N = animal numbers, other aspects of their data remained as-is. The description of the statistics in the figure legend is also dense and difficult to follow in places. Ns should be checked in the legend and figure to make sure they're correct, as at least one error was noted (e.g., see Fig. 2C). Overall, the authors' response falls short of the standard of rigor that helps to reinforce scientific findings from reliability and reproducibility concerns when generating more data to increase Ns (i.e., the number of animals) would have been the better choice.

Another persistent concern from the previous review is that, in the electron microscopic analyses of dendritic spines (Fig. 5), the authors only compared fear acquisition versus extinction training. One critique was that the lack of inclusion of a naïve control group made it difficult to understand how these structural synaptic changes are occurring relative to baseline. It was also noted that the authors appropriately included naïve controls in other experiments in the paper. In the revised submission the authors simply added in naïve control data to their previous histogram. It is not considered good practice to collect, process, or analyze data one group at a time, as this would be prone to cohort effects or experimental bias. These data should be discarded and the experiment should be run correctly with randomized cases in each group, or instead these data should be eliminated from the report since there is a key control group missing. Again, the nature of the authors' response perpetuates the aforementioned concern that data collection and analysis in this report may fall short of an acceptable standard of rigor.

---

## [Referee Report · Reviewer #2 (Public review)]

Summary:

Ziółkowska et al. characterize the synaptic mechanisms at the basis of the RE-dCA1 contribution to the consolidation of fear memory extinction. In particular, they describe a layer specific modulation of RE-dCA1 excitatory synapses modulation associated to contextual fear extinction which is impaired by transient chemogenetic inhibition of this pathway. These results indicate that RE activity-mediated modulation of synaptic morphology contributes to contextual fear extinction

Strengths:

The manuscript is well conceived, the statistical analysis is solid and methodology appropriate. The strength of this work is that it nicely builds up on existing literature and provides new molecular insight on a thalamo-hippocampal circuit previously known for its role in fear extinction. In addition, the quantification of pre- and post-synapses is particularly thorough.

Weaknesses:

The results illustrated in this manuscript show nice incremental evidence about the neural mechanisms contributing to the RE-CA1 modulation of fear extinction. The novelty of this manuscript is therefore not exceptional, but still highly relevant for the field.

---

## [Author Response]

The following is the authors’ response to the original reviews.

**Public Reviews:**

**Reviewer #1 (Public review):**
The findings of Ziolkowska and colleagues show that a specific projection from the nucleus reuniens of the thalamus (RE) to dorsal hippocampal CA1 neurons plays an important role in fear extinction learning in male and female mice. In and of itself, this is not a particularly new finding, although the authors' identification of structural alterations from within dorsal CA1 stratum lacunosum moleculare (SLM) as a candidate mechanism for the learning-related plasticity is potentially novel and exciting. The authors use a range of anatomical and functional approaches to demonstrate structural synaptic changes in dorsal CA1 that parallel the necessary role of RE inputs in modulating extinction learning. Yet, the significance of these findings is substantially limited by several technical shortcomings in the experimental design, and the authors' central interpretation. Otherwise, there remain several strengths in the design and interpretation that offset some of these concerns.Given that much is already known about the role of RE and hippocampus in modulating fear learning and extinction, it remains unclear whether addressing these concerns would substantially increase the impact of this study beyond the specific area of speciality. Below, several major weaknesses will be highlighted, followed by several miscellaneous comments.Methodological:(1) One major methodological weakness in the experimental design involves the widespread misapplication of Ns used for the statistical analyses. Much of the anatomical analyses of structural synaptic changes in the RE-CA1 pathway use N = number of axons (Figs. 1, 2), N = number of dendrites (Figs. 3, 4), and N = number of sections (Fig. 7; note that there are 7 figures in total). In every instance, N = animal number should be used. It is unclear which of these results would remain significant if N = animal number were used in each or how many more animals would be required. This is problematic since these data comprise the main evidence for the authors' central conclusion that specific structural synaptic changes are associated with fear extinction learning.

We do agree with the reviewer that N = animal number is the preferred way to present data in most of our experiments. However, in some experimental groups we observed a very low number of entries. For example, in the 5US group we found RE+/+ spines only in 3 out of 6 analyzed animals. We believe that this observation is not due to technical problems as mCherry virus transduction required to find RE+/+ spines is similar in all experimental groups and we analyzed similar volumes of tissue. While this result still allows the calculation of density of RE+/+ spines per animal it generates no entries for spine area and PSD95 mean gray value if N = animal number. Hence, we decided to use N=animals to calculate spines and boutons densities, and N=dendritic spines/boutons to calculate other spine/bouton parameters.

(2) There is a lack of specific information regarding what constitutes learning with respect to behavioral freezing. It is never clearly stated what specific intervals are used over which freezing is measured during acquisition, extinction, and in extinction retrieval tests. Additionally, assessment of freezing during retrieval at 5- and 30-min time points doesn't lay to rest the possibility that there were differences in the decay rate over the 30-min period (also see below).

We added a detailed description of how learning was assessed.

ln 125-134: “For assessment of learning we used percent of time spent by animals freezing (% freezing). Freezing behavior was defined as complete lack of movement, except respiration. To assess within-session learning (working memory) we compared pre- and post-US freezing frequency (the first 148 sec vs last 30 sec) during the CFC session (day 1). To assess formation of long-term contextual fear memory, we compared pre-US freezing (day 1) and the first 5 minutes of the Extinction session (day 2). To assess within session contextual fear extinction we ran 2-way ANOVA to assess the effect of time and manipulation on freezing frequency. Freezing data were analyzed in 5-minute bins. To assess formation of long-term contextual fear extinction memory we compared the first 5 minutes of the Extinction session (day 2) and Test session (day 3).”

As suggested by the reviewer, we also added data for all six 5-minut bins of Extinction sessions.

(3) A minor-to-moderate methodological weakness concerns the authors' decision to utilize saline injected groups as controls for the chemogenetics experiments (Figs. 5, 6). The correct design is to have a CNO-only group with the same viral procedure sans hM4Di. This concern is partly mitigated by the inclusion of a CNO vs. saline injection control experiment (Fig. 6).

Figure 5 does not describe a chemogenetic experiment.

We added new groups with control virus (CNO vs saline) to Figure 6 (now Fig. 6D and H).

The chemogenetic experiment shown on Figure 7 has all 4 experimental groups (Control vs hM4Di and saline vs CNO).

(4) In the electron microscopic analyses of dendritic spines (Fig. 5), comparison of only the fear acquisition versus extinction training, and the lack of inclusion of a naïve control group, makes it difficult to understand how these structural synaptic changes are occurring relative to baseline. It is noteworthy that the authors utilize the tripartite design in other anatomical analyses (Fig. 2-4).

We added data for the Naive mice to Figure 5.

(5) Interpretation:The main interpretive weakness in the study is the authors' claim that their data shows a role for the RE-CA1 pathway in memory consolidation (i.e., see Abstract). This claim is based on the premise that, although RE-CA1 pathway inactivation with CNO treatment 30 min prior to contextual fear extinction did not affect freezing at 5- and 30-min time points relative to saline controls, these rats showed greater freezing when tested on extinction retrieval 24 h thereafter. First, the data do not rule out possible differences in the decay rate of freezing during extinction training due to CNO administration. Next, the fact that CNO is given prior to training still leaves open the possibility that acquisition was affected, even if there were not any frank differences in freezing. Support for this latter possibility derives from the fact that mice tested for extinction retrieval as early as 5 min after extinction training (Fig. 6C) showed the same impairments as mice tested 24 h later (Figs. 6A). Further, all the structural synaptic changes argued to underlie consolidation were based on analysis at a time point immediately following extinction training, which is too early to allow for any long-term changes that would underlie memory consolidation, but instead would confer changes associated with the extinction training event.

We do agree with the reviewer that our data do not allow us to conclude whether RE-CA1 pathway is involved in acquisition or consolidation of CFE memory. Therefore, we avoid those terms in the manuscript. We just conclude that RE→CA1 participates in the CFE.

**Reviewer #2 (Public review):**
Summary:Ziółkowska et al. characterize the synaptic mechanisms at the basis of the REdCA1 contribution to the consolidation of fear memory extinction. In particular, they describe a layer specific modulation of RE-dCA1 excitatory synapses modulation associated to contextual fear extinction which is impaired by transient chemogenetic inhibition of this pathway. These results indicate that RE activity-mediated modulation of synaptic morphology contributes to the consolidation of contextual fear extinctionStrengths:The manuscript is well conceived, the statistical analysis is solid and methodology appropriate. The strength of this work is that it nicely builds up on existing literature and provides new molecular insight on a thalamo-hippocampal circuit previously known for its role in fear extinction. In addition, the quantification of pre- and post-synapses is particularly thorough.Weaknesses:The findings in this paper are well supported by the data more detailed description of the methods is needed.(1) In the paragraph Analysis of dCA1 synapses after contextual fear extinction (CFE), more experimental and methodological data should be given in the text:- how was PSD95 used for the analysis, what was the difference between RE. Even if Thy1-GFP mice were used in Fig.2, it appears they were not used for bouton size analysis. To improve clarity, I suggest moving panel 2C to Figure 3. It is not clear whether all RE axons were indiscriminately analysed in Fig. 2 or if only the ones displaying colocalization with both PSD95 and GFP were analysed. If GFP was not taken into account here, analysed boutons could reflect synapses onto inhibitory neurons and this potential scenario should be discussed.

PSD-95 immunostaining in close apposition to boutons was used to identify RE buttons innervating CA1 (Fig 1 and 2). In these cases PSD-95 signal was not quantified. PSD-95 in close apposition to dendritic spines was used as a proxy of PSDs in CA1 (Figure 3, 4 and 7). In these cases we assessed the integrated mean gray value of PSD-95 signal per dendritic spine (Figure 3, 4) or per ROI (Figure 7). This is explained in detail in the section Confocal microscopy and image quantification (ln 149-172).

GFP signal was not taken into account during boutons analysis. This is explained in the materials and methods section Confocal microscopy and image quantification (ln 149-172).

We indicate that PSD-95 is a marker of excitatory synapses located both on excitatory and inhibitory neurons.

Ln 258: RE boutons were identified in SO and SLM as axonal thickenings in close apposition to PSD-95-positive puncta a synaptic scaffold used as a marker of excitatory synapses located both on excitatory and inhibitory neurons (Kornau et al., 1995; El-Husseini et al., 2000; Chen et al., 2011; Dharmasri et al., 2024).

We also cite literature demonstrating that RE projects to the hippocampal formation and forms asymmetric synapses with dendritic spines and dendrites, suggesting innervation of excitatory synapses on both excitatory and aspiny inhibitory neurons (ln 673).

As advised by the reviewer the Figure 2C panel was moved to Figure 3 (now it is Fig 3A).

(2) in the methods: The volume of intra-hippocampal CNO injections should be indicated. The concentration of 3 uM seems pretty low in comparison with previous studies. CNO source is missing.

This section has been rewritten to be more clear. The concentration of CNO was chosen based on the previous studies (Stachniak et al., 2014).

ln 103: “Cannula placement. Mice were anesthetized by inhalation of 3–5% isoflurane (IsoFlo; Abbott Animal Health) in oxygen and positioned in a stereotaxic frame (51503, Stoelting, Wood Dale, IL, USA). Two holes were drilled in the skull, and a double guide cannulae (2 mm apart and 2 mm long; 26GA, Plastics One) was lowered into the holes such that the cannula tip was located over dorsal CA1 area (2 mm posterior to bregma, ±1 mm lateral, and −1.3 mm vertical). Cannulae were kept patent by using 33-gauge internal dummy cannulae (Plastics One). The animals were used in contextual fear conditioning 21 days after the cannulation. Animals received bilateral CNO (3 μM, 0.2 μl per side for 1 min; Tocris Bioscience, Cat. No. 4936) (Stachniak et al., 2014) or saline injections (0.2 μl per side) 30 minutes before Extinction session via intrahippocampal injection cannulae (33-gauge). After the infusion, the cannula was left in place for 30 seconds. The cannula placement was verified by histology, and only data from animals with correct cannula implants were included in statistical analyses.”

(3) More details of what software/algorithm was used to score freezing should be included.

Freezing was automatically scored with VideoFreeze Software (Med Associates Inc).

(4) Antibody dilutions for IHC should be indicated. Secondary antibody incubation time should be indicated.

The missing information is added.

ln 144: “Next, sections were incubated in 4°C overnight with primary antibodies directed against PSD-95 (1:500, Millipore, MAB 1598), washed three times in 0.3% Triton X-100 in PBS and incubated in room temperature for 90 minutes with a secondary antibody bound with Alexa Fluor 647 (1:500, Invitrogen, A31571).”

(5) No statement about code and data availability is present.

The statements are added.

ln 785: Row data and the code used for analysis of confocal data is available at OSF (https://osf.io/bnkpx/).

**Reviewer #3 (Public review):**
Summary:This paper examined the role of nucleus reuniens (RE) projections to dorsal CA1 neurons in context fear extinction learning. First, they show that RE neurons send excitatory projections to the stratum oriens (SO) and the stratum lacunosum moleculare (SLM), but not the stratum radiatum (SR). After context fear conditioning, the synaptic connections between RE and dCA1 neurons in the SLM (but not the SO) are weakened (reduced bouton and spine density) after mice undergo context fear conditioning. This weakening is reversed by extinction learning, which leads to enhanced synaptic connectivity between RE inputs and dendrites in the SLM. Control experiments demonstrate that the observed changes are due to extinction and not caused by simple exposure to the context. Extinction learning also induced increases in the size (volume and surface area) of the post-synaptic density (PSD) in SLM. To establish the functional role of RE inputs to dCA1, the researchers used an inhibitory DREADD to silence this pathway during extinction learning. They observe that extinction memory (measured 2-hours or 24-hours later) is impaired by this inhibition. Control experiments show that the extinction memory deficit is not simply due to increased freezing caused by inactivation of the pathway or injections of CNO. Inhibiting the RO projection during extinction learning also reduced the levels of PSD-95 protein levels in the spines of dCA1 neurons.Strengths:Based on their results, the authors conclude that, "the RE→SLM pathway participates in the updating of fearful context value by actively regulating CFE-induced molecular and structural synaptic plasticity in the SLM.". I believe the data are generally consistent with this hypothesis, although there is an important control condition missing from the behavioral experiments.Weaknesses:(1) A defining feature of extinction learning is that it is context specific (Bouton, 2004). It is expressed where it was learned, but not in other environments. Similarly, it has been shown that internal contexts (or states) also modulate the expression of extinction (Bouton, 1990). For example, if a drug is administered during extinction learning, it can induce a specific internal state. If this state is not present during subsequent testing, the expression of extinction is impaired just as it is when the physical context is altered (Bouton, 2004). It is possible that something similar is happening in Figure 6. In these experiments, CNO is administered to inactivate the RE-dCA1 projection during extinction learning. The authors observe that this manipulation impairs the expression of extinction the next day (or 2-hours later). However, the drug is not given again during the test. Therefore, it is possible that CNO (and/or inactivation of the RE-dCA1 pathway) induces a state change during extinction that is not present during subsequent testing. Based on the literature cited above, this would be expected to disrupt fear extinction as the authors observed. To determine if this alternative explanation is correct, the researchers need to add groups that receive CNO during extinction training and subsequent extinction testing. If the deficits in extinction expression reported in Figure 6 result from a state change, then these groups should not exhibit an impairment. In contrast, if the authors' account is correct, then the expression of extinction should still be disrupted in mice that receive CNO during training and testing.

We do agree with the reviewer that such an experiment would be interesting. However, it could be also confusing as we could not distinguish whether the possible behavioral effects are related to the state-dependent aspects of CFE or impaired recall of CFE. Importantly, previous studies showed that RE is crucial for extinction recall (Totty et al., 2023). We also show that CFE memory is impaired not only when the animals recall CFE without CNO (day 3) but also with CNO (day 4) (Figure 6C). Moreover, we do not see the effects of CNO on CFE in the control groups (Figure 6D and H). So we believe that it is unlikely that CNO results in state-dependent CFE.

(2) In their analysis of dCA1 synapses after contextual fear extinction (CFE) (Figure 4), the authors should have compared Ctx and Ctx-Ctx animals against naïve animals (as they did in Figure 3) when comparing 5US and Ext with naïve animals. Otherwise, the authors cannot make the following conclusion; "since changes of SLM synapses were not observed in the animals exposed to the familiar context that was not associated with the USs, our data support the role of the described structural plasticity at the RE→SLM synapses in CFE, rather than in processing contextual information in general.".

We assume that the key experimental groups to conclude about synaptic plasticity related to particular behavior are the groups that differ just by one factor/experience. For CFE that would be mice sacrificed immediately before and after CFE session (Figure 2 & 3); on the other hand to conclude about the effects of the re-exposure to the neutral context mice sacrificed before and after second exposure to the neutral context are needed (Figure 4). The naive group, as it differs by at least two manipulations from the Ext and Ctx-Ctx groups, is interesting but not crucial in both cases. This group would be necessary if we focused on the memories of FC or novel context. However, these topics are not the main focus of the current manuscript. Still, the naive group is shown on Figures 2 & 3 to check if CFE brings spine parameters to the levels observed in mice with low freezing.

We have re-written the cited paragraph to be more precise in our conclusions.

"Overall, our data demonstrate that synapses in all dCA1 strata undergo structural or molecular changes relevant to CFC and/or CFE. However, only in SLM CFE-induced synaptic changes are likely to be directly regulated by RE inputs as they appear on RE+ dendrites and spines. Since such changes of SLM synapses were not observed in the animals re-exposed to the neutral context, our data support the role of the described structural plasticity at the RE→SLM synapses in CFE, rather than in processing contextual information in general."

(3) In the materials and methods section, the description of cannula placements is confusing and needs to be rewritten.

This section has been rewritten.

ln 103: “Cannula placement. Mice were anesthetized by inhalation of 3–5% isoflurane (IsoFlo; Abbott Animal Health) in oxygen and positioned in a stereotaxic frame (51503, Stoelting, Wood Dale, IL, USA). Two holes were drilled in the skull, and a double guide cannulae (2 mm apart and 2 mm long; 26GA, Plastics One) was lowered into the holes such that the cannula tip was located over dorsal CA1 area (2 mm posterior to bregma, ±1 mm lateral, and −1.3 mm vertical). Cannulae were kept patent by using 33-gauge internal dummy cannulae (Plastics One). The animals were used in contextual fear conditioning 21 days after the cannulation. Animals received bilateral CNO (3 μM, 0.2 μl per side for 1 min; Tocris Bioscience, Cat. No. 4936) (Stachniak et al., 2014) or saline injections (0.2 μl per side) 30 minutes before Extinction session via intrahippocampal injection cannulae (33-gauge). After the infusion, the cannula was left in place for 30 seconds. The cannula placement was verified by histology, and only data from animals with correct cannula implants were included in statistical analyses.”

**Recommendations for the authors:**

**Reviewer #1 (Recommendations for the authors):**
Other/ Minor:In the beginning of the second paragraph on p. 21 of the Results section, it states that "RE-dCA1 has no effect on working memory," although it was not clear what data the authors were referring to support this conclusion.

We refer there to the changes of freezing behavior within the CFE session. This is explained now.

**Reviewer #2 (Recommendations for the authors):**
No statement about code and data availability is present.

The statements are added.

ln 785: “Row data and the code used for analysis of confocal data is available at OSF (https://osf.io/bnkpx/).”